# MODEL INVERSION NETWORKS FOR MODEL-BASED OPTIMIZATION

## ABSTRACT

In this work, we aim to solve data-driven optimization problems, where the goal is to find an input that maximizes an unknown score function given access to a dataset of input, score pairs. Inputs may lie on extremely thin manifolds in high-dimensional spaces, making the optimization prone to falling-off the manifold. Further, evaluating the unknown function may be expensive, so the algorithm should be able to exploit static, offline data. We propose *model inversion networks* (MINs) as an approach to solve such problems. Unlike prior work, MINs scale to extremely high-dimensional input spaces and can efficiently leverage offline logged datasets for optimization in both contextual and non-contextual settings. We show that MINs can also be extended to the active setting, commonly studied in prior work, via a simple, novel and effective scheme for active data collection. Our experiments show that MINs act as powerful optimizers on a range of contextual/non-contextual, static/active problems including optimization over images and protein designs and learning from logged bandit feedback.

## 1 INTRODUCTION

Data-driven optimization problems arise in a range of domains: from protein design (Brookes et al., 2019) to automated aircraft design (Hoburg & Abbeel, 2012), from the design of robots (Liao et al., 2019) to the design of neural net architectures (Zoph & Le, 2017) and learning from logged feedback, such as optimizing user preferences in recommender systems. Such problems require optimizing unknown reward or score functions using previously collected data consisting of pairs of inputs and corresponding score values, without direct access to the score function being optimized. This can be especially challenging when valid inputs lie on a low-dimensional manifold in the space of all inputs, e.g., the space of valid aircraft designs or valid images. Existing methods to solve such problems often use derivative-free optimization (Snoek et al.). Most of these techniques require *active* data collection where the unknown function is queried at new inputs. However, when function evaluation involves a complex real-world process, such as testing a new aircraft design or evaluating a new protein, such active methods can be very expensive. On the other hand, in many cases there is considerable prior data – existing aircraft and protein designs, and advertisements and user click rates, etc. – that could be leveraged to solve the optimization problem.

In this work, our goal is to develop an optimization approach to solve such optimization problems that can (1) readily operate on high-dimensional inputs comprising a narrow, low-dimensional manifold, such as natural images, (2) readily utilize offline static data, and (3) learn with minimal active data collection if needed. We can define this problem setting formally as the optimization problem

$$\mathbf{x}^{\star} = \arg\max_{\mathbf{x}} f(\mathbf{x}), \tag{1}$$

where the function $f(\mathbf{x})$ is unknown, and we have access to a dataset $\mathcal{D} = \{(\mathbf{x}_1, y_1), \ldots, (\mathbf{x}_N, y_N)\}$, where $y_i$ denotes the value $f(\mathbf{x}_i)$. If no further data collection is possible, we call this the data-driven model-based optimization setting. This can also be extended to the *contextual* setting, where the aim is to optimize the expected score function value across a context distribution. That is,

$$\pi^{\star} = \arg\max_{\pi} \mathbb{E}_{c \sim p_0(\cdot)}[f(c, \pi(c))], \tag{2}$$

where $\pi^{\star}$ maps contexts $c$ to inputs $\mathbf{x}$, such that the expected score under the context distribution $p_0(c)$ is optimized. As before, $f(c, \mathbf{x})$ is unknown and we have access to a dataset $\mathcal{D} = \{(c_i, \mathbf{x}_i, y_i)\}_{i=1}^{N}$,

where $y_i$ is the value of $f(c_i, \mathbf{x}_i)$. Such contextual problems with logged datasets have been studied in the context of contextual bandits (Swaminathan & Joachims, a; Joachims et al., 2018).

A simple way to approach these model-based optimization problems is to train a proxy function $f_\theta(\mathbf{x})$ or $f_\theta(c, \mathbf{x})$, with parameters $\theta$, to approximate the true score, using the dataset $\mathcal{D}$. However, directly using $f_\theta(\mathbf{x})$ in place of the true function $f(\mathbf{x})$ in Equation (1) generally works poorly, because the optimizer will quickly find an input $\mathbf{x}$ for which $f_\theta(\mathbf{x})$ outputs an erroneously large value. This issue is especially severe when the inputs $\mathbf{x}$ lie on a narrow manifold in a high-dimensional space, such as the set of natural images (Zhu et al., 2016). The function $f_\theta(\mathbf{x})$ is only valid near the training distribution, and can output erroneously large values when queried at points chosen by the optimizer. Prior work has sought to addresses this issue by using uncertainty estimation and Bayesian models (Snoek et al., 2015) for $f_\theta(\mathbf{x})$, as well as active data collection (Snoek et al.). However, explicit uncertainty estimation is difficult when the function $f_\theta(\mathbf{x})$ is very complex or when $\mathbf{x}$ is high-dimensional.

Instead of learning $f_\theta(\mathbf{x})$, we propose to learn the inverse function, mapping from values $y$ to corresponding inputs $\mathbf{x}$. This inverse mapping is one-to-many, and therefore requires a *stochastic* mapping, which we can express as $f_\theta^{-1}(y, \mathbf{z}) \rightarrow \mathbf{x}$, where $\mathbf{z}$ is a random variable. We term such models *model inversion networks* (MINs). MINs provide us with a number of desirable properties: they can utilize static datasets, handle high-dimensional input spaces such as images, can handle contextual problems, and can accommodate both static datasets and active data collection. We discuss how to design simple active data collection methods for MINs, leverage advances in deep generative modeling (Goodfellow et al.; Brock et al., 2019), and scale to very high-dimensional input spaces. We experimentally demonstrate MINs in a range of settings, showing that they outperform prior methods on high-dimensional input spaces, perform competitively to Bayesian optimization methods on tasks with active data collection and lower-dimensional inputs, and substantially outperform prior methods on contextual optimization from logged data (Swaminathan & Joachims, a).

## 2 RELATED WORK

**Bayesian optimization.** In this paper, we aim to solve data-driven optimization problems. Most prior work aimed at solving such optimization problems has focused on the active setting. This includes algorithms such as the cross entropy method (CEM) and related derivative-free methods Rubinstein (1996); Rubinstein & Kroese (2004), reward weighted regression Peters & Schaal, Bayesian optimization methods based on Gaussian processes Shahriari et al. (2016); Snoek et al.; 2015), and variants that replace GPs with parametric acquisition function approximators such as Bayesian neural networks (Snoek et al., 2015) and latent variable models (Kim et al., 2019; Garnelo et al., 2018b;a), as well as more recent methods such as CbAS (Brookes et al., 2019). These methods require the ability to query the true function $f(\mathbf{x})$ at each iteration to iteratively arrive at a near-optimal solution. We show in Section 3.3 that MINs can be applied to such an active setting as well, and in our experiments we show that MINs can perform competitively with these prior methods. Additionally, we show that MINs can be applied to the static setting, where these prior methods are not applicable. Furthermore, most conventional BO methods do not scale favourably to high-dimensional input spaces, such as images, while MINs can handle image inputs effectively.

**Contextual bandits.** Equation 2 captures the class of contextual bandit problems. Prior work on batch contextual bandits has focused on batch learning from bandit feedback (BLBF), where the learner needs to produce the best possible policy that optimizes the score function from logged experience. Existing approaches build on the counterfactual risk minimization (CRM) principle (Swaminathan & Joachims, a;b), and have been extended to work with deep nets (Joachims et al., 2018). In our comparisons, we find that MINs substantially outperform these prior methods in the batch contextual bandit setting.

**Deep generative modeling.** Recently, deep generative modeling approaches have been very successful at modelling high-dimensional manifolds such as natural images (Goodfellow et al.; Van Den Oord et al.; Dinh et al., 2016), speech (van den Oord et al., 2018), text (Yu et al.), alloy composition prediction (Nguyen et al.), etc. MINs combine the strength of such generative models with important algorithmic decisions to solve model-based optimization problems. In our experimental evaluation, we show that these design decisions are important for adapting deep generative models to model-based optimization, and it is difficult to perform effective optimization without them.

## 3 Model Inversion Networks

In this section, we describe our model inversion networks (MINs) method, which can perform both active and passive model-based optimization over high-dimensional input spaces.

**Problem statement.** Our goal is to solve optimization problems of the form $\mathbf{x}^{\star} = \arg\max_{\mathbf{x}} f(\mathbf{x})$, where the function $f(\mathbf{x})$ is not known, but we must instead use a dataset of input-output tuples $\mathcal{D} = \{(\mathbf{x}_i, y_i)\}$. In the contextual setting described in Equation (2), each datapoint is also associated with a context $c_i$. For clarity, we present our method in the non-contextual setting, but the contextual setting can be derived analogously by conditioning all functions on the context. In the *active* setting, which is most often studied in prior work, the algorithm is allowed to actively query $f(\mathbf{x})$ one or more times on each iteration to augment the dataset, while in the *static* setting, only an initial static dataset is available. The goal is to obtain the best possible $\mathbf{x}^{\star}$ (i.e., the one with highest possible value of $f(\mathbf{x}^{\star})$).

One naïve way of solving MBO problems is to learn a proxy score function $f_{\theta}(\mathbf{x})$, via standard empirical risk minimization. We could then maximize this learned function with respect to $\mathbf{x}$ via standard optimization methods. However, naïve applications of such a method would fail for two reasons. First, the proxy function $f_{\theta}(\mathbf{x})$ may not be accurate outside the samples on which it is trained, and optimization with respect to it may simply lead to values of $\mathbf{x}$ for which $f_{\theta}(\mathbf{x})$ makes the largest mistake in the negative direction. The second problem is more subtle. When $\mathbf{x}$ lies on a narrow manifold in very high-dimensional space (such as the space of natural images), the optimizer can produce invalid values of $\mathbf{x}$, which result in arbitrary outputs when fed into $f_{\theta}(\mathbf{x})$. Since the shape of this manifold is unknown, it is difficult to constrain the optimizer to prevent this. This second problem is rarely addressed or discussed in prior work, which typically focuses on optimization over low-dimensional and compact domains with known bounds.

### 3.1 Optimization via Inverse Maps

Part of the reason for the brittleness of the naïve approach above is that $f_{\theta}(\mathbf{x})$ has a high-dimensional input space, making it easy for the optimizer to find inputs $\mathbf{x}$ for which the proxy function produces an unreasonable output. Can we instead learn a function with a small input space, which implicitly understands the space of valid, in-distribution values for $\mathbf{x}$? The main idea behind our approach is to model an inverse map that produces a value of $\mathbf{x}$ given a score value $y$, given by $f_{\theta}^{-1} : \mathcal{Y} \to \mathcal{X}$. The input to the inverse map is a scalar, making it comparatively easy to constrain to valid values, and by directly generating the inputs $\mathbf{x}$, an approximation to the inverse function must implicitly understand which input values are valid. As multiple $\mathbf{x}$ values can correspond to the same $y$, we design $f_{\theta}^{-1}$ as a stochastic map that maps a score value along with a $d_z$-dimensional random vector to a $\mathbf{x}$, $f_{\theta}^{-1} : \mathcal{Y} \times \mathcal{Z} \to \mathcal{X}$, where $\mathbf{z}$ is distributed according to a prior distribution $p_0(\mathbf{z})$.

To define the inverse map objective, let the data distribution be denoted $p_{\mathcal{D}}(\mathbf{x}, y)$, let $p_{\mathcal{D}}(y)$ be the marginal over $y$, and let $p(y)$ be an any distribution defined on $\mathcal{Y}$ (which could be equal to $p_{\mathcal{D}}(y)$). We can train the proxy inverse map $f_{\theta}^{-1}$ under distribution $p(y)$ by minimizing the following objective:

$$\mathcal{L}_p(\mathcal{D}) = \mathbb{E}_{y \sim p(y)}[D(p_{\mathcal{D}}(\mathbf{x}|y), p_{f_{\theta}^{-1}}(\mathbf{x}|y))], \qquad (3)$$

where $p_{f_{\theta}^{-1}}(\mathbf{x}|y)$ is obtained by marginalizing over $\mathbf{z}$, $p_{f_{\theta}^{-1}}(\mathbf{x}|y) = \int_{\mathbf{z}} p_0(\mathbf{z}) \cdot \mathbf{1}[[\mathbf{x} = f_{\theta}^{-1}(\mathbf{z}, y)]] d\mathbf{z}$, and $D$ is a measure of divergence between the two distributions. Using the Kullback-Leibler divergence leads to maximum likelihood learning, while Jensen-Shannon divergence motivates a GAN-style training objective. MINs can be adapted to the contextual setting by passing in the context as an input and learning $f_{\theta}^{-1}(y_i, z, c_i)$. In standard empirical risk minimization, we would choose $p(y)$ to be the data distribution $p_{\mathcal{D}}(y)$, such that the expectation can be approximated simply by sampling train-ing tuples $(\mathbf{x}_i, y_i)$ from the training set. How-ever, as we will discuss in Section 3.3, a more careful choice for $p(y)$ can lead to better perfor-mance. The MIN algorithm is based on training an inverse map, and then using it via the infer-ence procedure in Section 3.2 to infer the $\mathbf{x}$ that approximately optimizes $f(\mathbf{x})$. The structure of the MIN algorithm is shown in Algorithm 1.

---

**Algorithm 1** Generic Algorithm for MINs

1: Input: $p_{\mathcal{D}}(y)$: distribution of $y$ in $\mathcal{D}$
2: Train inverse map $f_{\theta}^{-1} : \mathcal{Y} \times \mathcal{Z} \to \mathcal{X}$ using objective (Equation 3) with reweighting, active data collection if needed
3: $\mathbf{x}^{\star} \leftarrow \text{Approx-Infer}(f_{\theta}^{-1}, p_{\mathcal{D}}(y))$
4: return $\mathbf{x}^{\star}$

---

## 3.2 INFERENCE WITH INVERSE MAPS (APPROX-INFER)

Once the inverse map is trained, the goal of our algorithm is to generate the best possible $\mathbf{x}^\star$, which will maximize the true score function as well as possible under the dataset. Since a score $y$ needs to be provided as input to the inverse map, we must select for which score $y$ to query the inverse map to obtain a near-optimal $\mathbf{x}$. One naïve heuristic is to pick the best $y_{\max} \in \mathcal{D}$ and produce $\mathbf{x}_{\max} \sim f_\theta^{-1}(y_{\max}^*)$ as the output. However, the method should be able to extrapolate beyond the best score seen in the dataset, especially in contextual settings, where a good score may not have been observed for all contexts.

In order to extrapolate as far as possible, while still staying on the valid data manifold, we need to measure the validity of the generated values of $\mathbf{x}$. One way to do this is to measure the agreement between the learned inverse map and an independently trained forward model $f_\theta$: the values of $y$ for which the generated samples $\mathbf{x}$ are predicted to have a score similar to $y$ are likely in-distribution, whereas those where the forward model predicts a very different score may be too far outside the training distribution. Since the latent variable $\mathbf{z}$ captures the multiple possible outputs of the one-to-many inverse map, we can further optimize over $\mathbf{z}$ for a given $y$ to find the best, most trustworthy output $\mathbf{x}$. This can be formalized as the following optimization:

$$\tilde{y}^*, \tilde{z}^* := \arg\max_{y,z} f_\theta(f_\theta^{-1}(z,y)) - \lambda_1 ||y - f_\theta(f_\theta^{-1}(z,y))||_2 + \lambda_2 \log p_0(z) \tag{4}$$

This optimization can be motivated as finding an extrapolated score that corresponds to values of $\mathbf{x}$ that lie on the valid input manifold, and for which independently trained forward and inverse maps agree. Although this optimization uses an approximate forward map $f_\theta(\mathbf{x})$, we show in our experiments in Section 4 that it produces substantially better results than optimizing with respect to a forward model alone. The inverse map substantially constraints the search space, requiring an optimization over a 1-dimensional $y$ and a (relatively) low-dimensional $\mathbf{z}$, rather than the full space of inputs. This scheme can be viewed as a special (deterministic) case of a probabilistic optimization procedure described in Appendix A.

## 3.3 REWEIGHTING THE TRAINING DISTRIBUTION

A naïve implementation of the training objective in Equation (3) samples $y$ from the data distribution $p_\mathcal{D}(y)$. However, as we are most interested in the inverse map's predictions for *high* values of $y$, it is much less important for the inverse map to predict accurate $\mathbf{x}$ values for values of $y$ that are far from the optimum. We could consider increasing the weights on datapoints with larger values of $y$. In the extreme case, we could train only on the best datapoint – either the single datapoint with the largest $y$ or, in the contextual case, the datapoint with the largest $y$ for each context. More generally, we can define the *optimal $y$ distribution* $p^*(y)$, which is simply the delta function centered on the best $y$, $p^*(y) = \delta_{y*}(y)$, in the deterministic case. If we instead assume that the observed scores have additive noise (i.e., we observe $f(\mathbf{x}) + \varepsilon, \varepsilon \sim \mathcal{N}$), then $p^*(y)$ would be a distribution centered around the optimal $y$. Of course, training on $p^*(y)$ is not practical, since it heavily down-weights most of the training data, leading to a very high-variance training objective, and is not even known in general, since the optimal data point is likely not in our training set. In this section, we will propose a better choice for $p(y)$ that trades off the variance due to an overly peaked training distribution and the bias due to training on the "wrong" distribution (i.e., anything other than $p^*(y)$).

We can train under a distribution other than the empirical distribution by using importance sampling, such that we sample from $p_\mathcal{D}$ and assign an importance weight, given by $\mathbf{w}_i = \frac{p(y_i)}{p_\mathcal{D}(y_i)}$, to each datapoint $(\mathbf{x}_i, y_i)$, where $p(y_i)$ is our desired distribution. The reweighted objective is given by $\hat{\mathcal{L}}_p(\mathcal{D}) := \frac{1}{|\mathcal{D}|} \sum_i \mathbf{w}_i \cdot \hat{D}(\mathbf{x}_i, f_\theta^{-1}(y_i))$. By bounding the variance and the bias of the gradient of $\hat{\mathcal{L}}_p(\mathcal{D})$ estimate, with respect to the reweighted objective without sampling error under $y$ drawn from $p^*(y)$, we obtain the following result: (Proof in Appendix B)

**Theorem 3.1** ((Informal) Bias + variance bound in MINs). *Let $\mathcal{L}(p^*)$ be the objective under $p^*(y)$ without sampling error: $\mathcal{L}(p^*) = \mathbb{E}_{y \sim p^*(y)}[D(p(\mathbf{x}|y), f^{-1}(y))]$. Let $N_y$ be the number of datapoints with the particular $y$ value observed in $\mathcal{D}$, For some constants $C_1, C_2, C_3$, with high confidence,*

$$\mathbb{E}\left[||\nabla_\theta \hat{\mathcal{L}}_p(\mathcal{D}) - \nabla_\theta \mathcal{L}(p^*)||_2^2\right] \leq C_1 \mathbb{E}_{y \sim p(y)}\left[\frac{1}{N_y}\right] + C_2 \frac{d_2(p||p_\mathcal{D})}{|\mathcal{D}|} + C_3 \cdot D_{\mathrm{TV}}(p^*, p)^2$$

Theorem 3.1 suggests a tradeoff between being close to the optimal distribution $p^*(y)$ and reducing variance by covering the full data distribution $p_\mathcal{D}$. We observe that the distribution $p(y)$ that minimizes

the RHS bound in Theorem 3.1 has the following form: $p(y) \propto \frac{N_y}{N_y+K} \cdot g(p^*(y))$, where $g(p^*)$ is a linear function of $p^*(y)$ that ensures that the distributions $p$ and $p^*$ are close. Theoretically, $g(\circ)$ is an increasing, piece-wise linear function of $\circ$. We can interpret the expression for $p(y)$ as a product of two likelihoods – the optimality of a particular $y$ value and the likelihood of a particular $y$ not being rare in $\mathcal{D}$. We empirically choose an exponential parameteric form for this function, which we describe in Section 3.5. This upweights the samples with higher scores, reduces the weight on *rare* $y$-values (i.e., those with low $N_y$), while preventing the weight on *common* $y$-values from growing, since $\frac{N_y}{N_y+K}$ saturates to 1 for large $N_y$. This is consistent with our intuition: we would like to upweight datapoints with high $y$-values, provided the number of samples at those values is not too low. Of course, for continuous-valued scores, we rarely see the same score twice. Therefore, we bin the $y$-values into discrete bins for the purpose of weighting, as we discuss in Section 3.5.

### 3.4 ACTIVE DATA COLLECTION VIA RANDOMIZED LABELING

While the passive setting requires care in finding the best value of $y$ for the inverse map, the active setting presents a different challenge: choosing a new query point $\mathbf{x}$ at each iteration to augment the dataset $\mathcal{D}$ and make it possible to find the best possible optimum. Prior work on bandits and Bayesian optimization often uses Thompson sampling (TS) (Russo & Van Roy, 2016; Russo et al., 2018; Srinivas et al.) as the data-collection strategy. TS maintains a posterior distribution over functions $p(f_t|\mathcal{D}_{1:t})$. At each iteration, it samples a function from this distribution and queries the point $\mathbf{x}_t^\star$ that greedily minimizes this function. TS offers an appealing query mechanism, since it achieves sub-linear Bayesian regret (defined as the expected cumulative difference between the value of the optimal input and the selected input), given by $\mathcal{O}(\sqrt{T})$, where $T$ is the number of queries.

Maintaining a posterior over high-dimensional parametric functions is generally intractable. However, we can devise a scheme to approximate Thompson sampling with MINs. To derive this method, first note that sampling $f_t$ from the posterior is equivalent to sampling $(\mathbf{x}, y)$ pairs consistent with $f_t$ – given sufficiently many $(\mathbf{x}, y)$ pairs, there is a unique smooth function $f_t$ that satisfies $y_i = f_t(\mathbf{x}_i)$. For example, we can infer a quadratic function exactly from three points. For a more formal description, we refer readers to the notion of Eluder dimension (Russo & Van Roy). Thus, instead of maintaining intractable beliefs over the function, we identify a function by the samples it generates, and define a way to sample synthetic $(\mathbf{x}, y)$ points such that they implicitly define a unique function sample from the posterior.

To apply this idea to MINs, we train the inverse map $f_{\theta_t}^{-1}$ at each iteration $t$ with an *augmented* dataset $\mathcal{D}_t' = \mathcal{D}_t \cup \mathcal{S}_t$, where $\mathcal{S}_t = \{(\tilde{\mathbf{x}}_j, \tilde{y}_j)\}_{j=1}^K$ is a dataset of synthetically generated input-score pairs corresponding to unseen $y$ values in $\mathcal{D}_t$. Training $f_{\theta_t}^{-1}$ on $\mathcal{D}_t'$ corresponds to training $f_{\theta_t}^{-1}$ to be an approximate inverse map for a function $f_t$ sampled from $p(f_t|\mathcal{D}_{1:t})$, as the synthetically generated samples $\mathcal{S}_t$ implicitly induce a model of $f_t$. We can then approximate Thompson sampling by obtaining $\mathbf{x}_t^\star$ from $f_{\theta_t}^{-1}$, labeling it via the true function, and adding it to $\mathcal{D}_t$ to produce $\mathcal{D}_{t+1}$. Pseudocode for this method, which we call "randomized labeling," is presented in Algorithm 2. In Appendix C, we further derive $\mathcal{O}(\sqrt{T})$ regret guarantees under mild assumptions. Implementation-wise, this method is simple, does not require estimating explicit uncertainty, and works with arbitrary function classes, including deep neural networks.

---

**Algorithm 2** Active Data Collection with Model Inversion Networks via Randomized Labeling

1: Initialize inverse map, $f_\theta^{-1} : \mathcal{Y} \times \mathcal{Z} \to \mathcal{X}$, dataset $\mathcal{D}_0 = \{\}$,
2: **for** step $t$ in $\{0, \ldots, \text{T-1}\}$ **do**
3:     Sample synthetic samples $\mathcal{S}_t = \{(\mathbf{x}_i, y_i)\}_{i=1}^K$ corresponding to unseen data points $y_i$ (by randomly pairing noisy observed $\mathbf{x}_i$ values with unobserved $y$ values.)
4:     Train *inverse map* $f_t^{-1}$ on $\mathcal{D}_t' = \mathcal{D}_t \cup \mathcal{S}_t$, using reweighting described in Section 3.3.
5:     Query function $f$ at $\mathbf{x}_t = f_t^{-1}(\max_{\mathcal{D}_t'} y)$
6:     Observe outcome: $(\mathbf{x}_t, f(\mathbf{x}_t))$ and update $\mathcal{D}_{t+1} = \mathcal{D}_t \cup (\mathbf{x}_t, f(\mathbf{x}_t))$
7: **end for**

---

### 3.5 PRACTICAL IMPLEMENTATION OF MINS

In this section, we describe our instantiation of MINs for high-dimensional inputs with deep neural network models. GANs (Goodfellow et al.) have been successfully used to model the manifold of

high-dimensional inputs, without the need for explicit density modelling and are known to produce more realistic samples than other models such as VAEs (Kingma & Welling, 2013) or Flows (Dinh et al., 2016). The inverse map in MINs needs to model the manifold of valid $\mathbf{x}$ thus making GANs a suitable choice. We can instantiate our inverse map with a GAN by choosing $D$ in Equation 3 to be the *Jensen-Shannon* divergence measure. Since we generate $\mathbf{x}$ conditioned on $y$, the discriminator is parameterized as $\mathrm{Disc}(\mathbf{x}|y)$, and trained to output 1 for a valid $(\mathbf{x}, y)$ pair (i.e., where $y = f(\mathbf{x})$ and $\mathbf{x}$ comes from the data) and 0 otherwise. Thus, we optimize the following objective:

$$\min_{\theta} \max_{\mathrm{Disc}} \mathcal{L}_p(\mathcal{D}) = \mathbb{E}_{y \sim p(y)}\big[\mathbb{E}_{\mathbf{x} \sim p_{\mathcal{D}}(\mathbf{x}|y)}[\log \mathrm{Disc}(x|y)] + \mathbb{E}_{\mathbf{z} \sim p_0(\mathbf{z})}[\log(1 - \mathrm{Disc}(f_{\theta}^{-1}(\mathbf{z}, y)|y))]\big]$$

This model is similar to a conditional GAN (cGAN), which has been used in the context of modeling distribution of $\mathbf{x}$ conditioned on a discrete-valued label (Mirza & Osindero, 2014). As discussed in Section 3.3, we additionally reweight the data distribution using importance sampling. To that end, we discretize the space $\mathcal{Y}$ into $B$ discrete bins $b_1, \cdots, b_B$ and, following Section 3.3, weight each bin $b_i$ according to $p(b_i) \propto \frac{N_{b_i}}{N_{b_i} + \lambda} \exp\left(\frac{|b_i - y^*|}{\tau}\right)$, where $N_{b_i}$ is the number of datapoints in the bin, $y^*$ is the maximum score observed, and $\tau$ is a hyperparameter. (After discretization, using notation from Section 3.3, for any $y$ that lies in bin $b$, $p^*(y) := p^*(b) = \exp\left(\frac{|b - y^*|}{\tau}\right)$ and $p(y) := p(b) \propto \frac{N_b}{N_b + \lambda} \exp\left(\frac{|b - y^*|}{\tau}\right)$.) Experimental details are provided in Appendix C.4.

In the active setting, we perform active data collection using the synthetic relabelling algorithm described in Section 3.4. In practice, we train two copies of $f_{\theta}^{-1}$. The first, which we call the exploration model $f_{\mathrm{expl}}^{-1}$, is trained with data augmented via synthetically generated samples (i.e., $\mathcal{D}'_t$). The other copy, called the exploitation model $f_{\mathrm{exploit}}^{-1}$, is trained on only real samples (i.e., $\mathcal{D}_t$). This improves stability during training, while still performing data collection as dictated by Algorithm 2. To generate the augmented dataset $\mathcal{D}'_t$ in practice, we sample $y$ values from $p^*(y)$ (the distribution over high-scoring $y$s observed in $\mathcal{D}_t$), and add positive-valued noise, thus making the augmented $y$ values higher than those in the dataset which promotes exploration. The corresponding inputs $x$ are simply sampled from the dataset $\mathcal{D}_t$ or uniformly sampled from the bounded input domain when provided in the problem statement. (for example, benchmark function optimization) After training, we infer best possible $\mathbf{x}^{\star}$ from the trained model using the inference procedure described in Section 3.2. In the active setting, the inference procedure is applied on $f_{\mathrm{exploit}}^{-1}$, the inverse map that is trained only on real data points.

## 4 EXPERIMENTAL EVALUATION

The goal of our empirical evaluation is to answer the following questions. **(1)** Can MINs successfully solve optimization problems of the form shown in Equations 1 and 2, in static settings and active settings, better than or comparable to prior methods? **(2)** Can MINs generalize to high dimensional spaces, where valid inputs $\mathbf{x}$ lie on a lower-dimensional manifold, such as the space of natural images? **(3)** Is reweighting the data distribution important for effective data-driven model-based optimization? **(4)** Does our proposed inference procedure effectively discover valid inputs $\mathbf{x}$ with better values than any value seen in the dataset? **(5)** Does randomized labeling help in active data collection?

### 4.1 DATA-DRIVEN OPTIMIZATION WITH STATIC DATASETS

We first study the *data-driven* model-based optimization setting. This requires generating points that achieve a better function value than any point in the training set or, in the contextual setting, better than the policy that generated the dataset *for each context*. We evaluate our method on a batch contextual bandit task proposed in prior work (Joachims et al., 2018) and on a high-dimensional contextual image optimization task. We also evaluate our method on several non-contextual tasks that require optimizing over high-dimensional image inputs to evaluate a semantic score function, including hand-written characters and real-world photographs.

**Batch contextual bandits.** We first study the contextual optimization problem described in Equation 2. The goal is to learn a policy, purely from static data, that predicts the correct bandit arm $\mathbf{x}$ for each context $c$, such that the policy achieves a high overall score $f(c, \pi(c))$ on average across contexts drawn from a distribution $p_0(c)$. We follow the protocol set out by Joachims et al. (2018),

| Dataset & Type | BanditNet | BanditNet$^*$ | MIN w/o I | MIN (Ours) | MINs w/o R |
|---|---|---|---|---|---|
| MNIST (49% corr.) | $36.42 \pm 0.6$ | $-$ | $94.2 \pm 0.13$ | $\mathbf{95.0 \pm 0.16}$ | $95.0 \pm 0.21$ |
| MNIST (Uniform) | $9.94 \pm 0.0$ | $-$ | $92.21 \pm 0.22$ | $\mathbf{93.67 \pm 0.51}$ | $92.8 \pm 0.01$ |
| CIFAR-10 (49% corr.) | $42.13 \pm 2.35$ | $87.0$ | $91.35 \pm 0.87$ | $\mathbf{92.21 \pm 1.0}$ | $89.02 \pm 0.05$ |
| CIFAR-10 (Uniform) | $14.43 \pm 1.43$ | $-$ | $76.31 \pm 0.40$ | $\mathbf{77.12 \pm 0.54}$ | $74.87 \pm 0.12$ |

Table 1: Test accuracy on MNIST and CIFAR-10 with 50k bandit feedback training examples. BanditNet$^*$ is the result from Joachims et al. (2018), while the BanditNet column is our implementation; we were unable to replicate the performance from prior work (details in Appendix D). MINs outperform both BanditNet and BanditNet$^*$, both with and without the inference procedure in Section 3.2. MINs w/o reweighting perform at par with full MINs on MNIST, and slightly worse on CIFAR 10, while still outperforming the baseline.

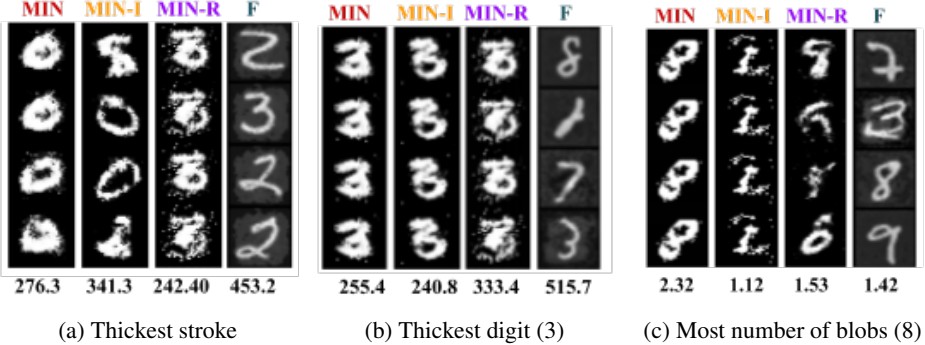

(a) Thickest stroke      (b) Thickest digit (3)      (c) Most number of blobs (8)

Figure 1: Results for non-contextual static dataset optimization on MNIST: (a) and (b): Stroke width optimization, and (c): Maximization of disconnected black pixel blobs. From left to right: MINs, MINs w/o Inference (Section 3.2), which sample $\mathbf{x}$ from the inverse map conditioned on the highest seen value of $y$, MINs w/o Reweighting (Section 3.3), and direct optimization of a forward model, which starts with a random dataset image and optimizes it for the highest score based on the forward model. Observe that MINs can produce thickest characters which resemble valid digits. Optimizing the forward function often turns non-digit pixels on, thus going off the valid manifold. Both the reweighting and inference procedure are important for good results. Quantitative results are provided in Appendix D.3. Different rows (for F) are obtained by optimizing from different initial points. Scores are listed beneath each figure. The larger score the better, provided the solution $\mathbf{x}$ is the image of a *valid* digit.

which evaluates contextual bandit policies trained on a static dataset for a simulated classification tasks. The data is constructed by selecting images from the (MNIST/CIFAR) dataset as the context $c$, a random label as the *input* $\mathbf{x}$, and a binary indicator indicating whether or not the label is correct as the *score* $y$. Multiple schemes can be used for selecting random labels for generating the dataset, and we evaluate on two such schemes, as described below. We report the average score on a set of new contexts, which is equal to the average 0-1 accuracy of the learned model on a held out test set of images (contexts). We compare our method to previously proposed techniques, including the BanditNet model proposed by Joachims et al. (2018) on the MNIST and CIFAR-10 (Krizhevsky, 2009) datasets. Note that this task is different from regular classification, in that the observed feedback $((c_i, \mathbf{x}_i, y_i)$ pairs) is partial, i.e. we do not observe the correct label for each context (image) $c_i$, but only whether or not the label in the training tuple is correct or not. We evaluate on two datasets: (1) data generated by selecting random labels $\mathbf{x}_i$ for each context $c_i$ and (2) data where the correct label is used 49% of the time, which matches the protocol in prior work (Joachims et al., 2018). We compare to BanditNet (Joachims et al., 2018) on identical dataset splits. We report the average 0-1 test accuracy for all methods in Table 1. The results show that MINs drastically outperform BanditNet on both MNIST and CIFAR-10 datasets, indicating that MINs can successfully perform contextual model-based optimization in the static (data-driven) setting. The results also show that utilizing the inference procedure in Section 3.2 produces an improvement of about 1.5% and 1.0% in test-accuracy on MNIST and CIFAR-10, respectively.

**Character stroke width optimization.** In the next experiment, we study how well MINs optimize over high-dimensional inputs, where valid inputs lie on a lower-dimensional manifold. We constructed an image optimization task out of the MNIST (LeCun & Cortes, 2010) dataset. The goal is to optimize *directly* over the image pixels, to produce images with the thickest stroke width, such that the image corresponds either (a) to any valid character or (b) a valid instance of a particular character class. A

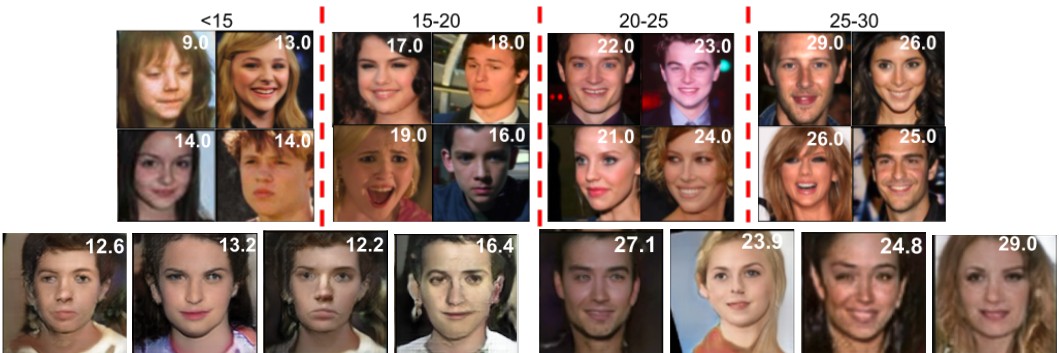

(a) Optimized **x** (trained on > 15 years)          (b) Optimized **x** (trained on > 25 years)

Figure 2: MIN optimization to obtain the youngest faces when trained on faces older than 15 (left) and older than 25 (right). Generated faces (bottom) are obtained via inference in the inverse map at different points during model training. Real faces of varying ages (including ages lower than those used to train the model) are shown in the top rows. We overlay the actual age (negative of the score function) for each face on the real images, and the age obtained from subjective user rankings on the generated faces. The score function being optimized (maximized) in this case is the negative age of the face.

successful algorithm will produce the thickest character that is still recognizable. In Figure 1, we observe that MINs generate images **x** that maximize the respective score functions in each case. We also evaluate on a harder task where the goal is to maximize the number of disconnected blobs of black pixels in an image of a digit. For comparison, we evaluate a method that directly optimizes the image pixels with respect to a forward model, of the form $f_\theta(\mathbf{x})$. In this case, the solutions are far off the manifold of valid characters. We also compare to MINs without the reweighting scheme and the inference procedure, where $y$ is the maximum possible $y$ in the dataset to demonstrate the benefits of these two aspects.

Table 2: Quantitative score-values for Youngest Face Optimization Task (larger the better)

| Task | MIN | MIN (best) |
|------|------|------------|
| $\geq 15$ | **-13.6** | -12.2 |
| $\geq 25$ | **-26.2** | -23.9 |

**Semantic image optimization.** The goal in these tasks is to quantify the ability of MINs to optimize high-level properties that require semantic understanding of images. We consider MBO tasks on the IMDB-Wiki faces (Rothe et al., 2015; 2016) dataset, where the function $f(\mathbf{x})$ is the negative of the age of the person in the image. Hence, images with younger people have higher scores.

We construct two versions of this task: one where the training data consists of all faces older than 15 years, and the other where the model is trained on all faces older than 25 years. This ensures that our model cannot simply copy the youngest face. To obtain ground truth scores for the generated faces, we use subjective judgement from human participants. We perform a study with 13 users. Each user was asked to answer a set of 35 binary-choice questions each asking the user to pick the older image of the two provided alternatives. We then fit an age function to this set of binary preferences, analogously to Christiano et al. (2017).

Figure 2 shows the images produced by MINs. For comparison, we also present some sample of images from the dataset partitioned by the ground truth score. We find that the most likely age for optimal images produced by training MINs on images of people 15 years or older was **13.6 years**, with the best image having an age of **12.2**. The model trained on ages 25 and above produced more mixed results, with an average age of **26.2**, and a minimum age of **23.9**. We report these results in Table 2. This task is exceptionally difficult, since the model must extrapolate outside of the ages seen in the training set, picking up on patterns in the images that can be used to produce faces that appear *younger* than any face that the model had seen, while avoiding unrealistic images.

We also conducted experiments on *contextual* image optimization with MINs. We studied contextual optimization over hand-written digits to maximize stroke width, using either the character category as the context $c$, or the top one-fourth or top half of the image. In the latter case, MINs must learn to complete the image while maximizing for the stroke width. In the case of class-conditioned optimization, MINs attain an average score over the classes of

Table 3: Quantitative score values for MNIST inpainting (contextual)

| Mask | MIN | Dataset |
|------|------|---------|
| *mask A* | **223.57** | 149.0 |
| *mask B* | **234.32** | 149.0 |

**237.6**, while the dataset average is **149.0**. In the case where the context is the top half or quarter of the image, MINs obtain average scores of **223.57** and **234.32**, respectively, while the dataset average is **149.0** for both tasks. We report these results in Table 3. We also conducted a contextual optimization experiment on faces from the Celeb-A dataset, with some example images shown in Figure 3. The context corresponds to the choice for the attributes brown hair, black hair, bangs, or moustache. The optimization score is given by the sum of the attributes wavy hair, eyeglasses, smiling, and no beard. Qualitatively, we can see that MINs successfully optimize the score while obeying the target context, though evaluating the true score is impossible without subjective judgement on this task. We discuss these experiments in more detail in Appendix D.1.

## 4.2 OPTIMIZATION WITH ACTIVE DATA COLLECTION

In the active MBO setting, MINs must select which *new* datapoints to query to improve their estimate of the optimal input. In this setting, we compare to prior model-based optimization methods, and evaluate the exploration technique described in Section 3.4.

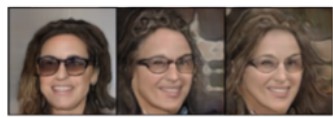

Brown or Black Hair

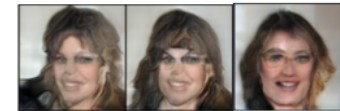

Both Brown/Black Hair

**Global optimization on benchmark functions.** We first compare MINs to prior work in Bayesian optimization on standard benchmark problems (DNGO) (Snoek et al., 2015): the 2D Branin function, and the 6D Hartmann function. As shown in Table 4, MINs reach within $\pm 0.1$ units of the global *minimum* (minimization is performed here, instead of maximization), performing comparably with commonly used Bayesian optimization methods based on Gaussian processes. We do not expect MINs to be as efficient as GP-based methods, since MINs rely on training parametric neural networks with many parameters, which is less efficient than GPs on low-dimensional tasks. Exact Gaussian processes and adaptive Bayesian linear regression (Snoek et al., 2015) outperform MINs in terms of optimization precision and the number of samples queried, but MINs achieve comparable performance with about $4\times$ more samples. We also report the performance of MINs without the random labeling exploration

Figure 3: Optimized $\mathbf{x}$ produced from contextual training on Celeb-A. Context = (brown hair, black hair, bangs, moustache and $f(\mathbf{x}) = \ell_1$(wavy hair, eyeglasses, smiling, no beard). We show the produced $\mathbf{x}^\star$ for two contexts. The model optimizes score for both observed contexts such as brown or black hair and extrapolates to unobserved contexts such as brown and black hair.

method, instead selecting the next query point by greedily maximizing the current model with some additive noise. We find that the random relabeling method produces substantially better results than the greedy data collection approach, indicating the importance of effective exploration methods for MINs.

| Function | Spearmint | DNGO | MIN | MIN + greedy |
|---|---|---|---|---|
| Branin (0.398) | $0.398 \pm 0.0$ | $0.398 \pm 0.0$ | $0.398 \pm 0.02$ | $0.4 \pm 0.05(800)$ |
| Hartmann6 (-3.322) | $-3.3166 \pm 0.02$ | $-3.319 \pm 0.00$ | $-3.315 \pm 0.05(600)$ | $-3.092 \pm 0.12(1200)$ |

Table 4: Active MBO on benchmark functions. The prior methods converge within 200 iterations. MINs require more iterations on some of the tasks, in which case we indicate the number of iterations in brackets. MINs reach similar final performance, and typically require 1-4$\times$ as much data as efficient GP-based algorithms.

**Protein fluorescence maximization.** In the next experiment, we study a high-dimensional active MBO task, previously studied by Brookes et al. (2019). This task requires optimizing over protein designs by selecting variable length sequences of codons, where each codon can take on one of 20 values. In order to model discrete values, we use a Gumbel-softmax GAN also previously employed in (Gupta & Zou, 2018), and as a baseline in (Brookes et al., 2019). For backpropagation, we choose a temperature $\tau = 0.75$ for the Gumbel-softmax operation. This is also mentioned in Appendix D. The aim in this task is to produce a protein with maximum fluorescence. Each algorithm is provided with a starting dataset, and then allowed a identical, limited number of score function queries. For each query made by an algorithm, it receives a score value from an oracle. We use the trained oracles released by Brookes et al. (2019). These oracles are separately trained forward models, and can potentially be inaccurate, especially for datapoints not observed in the starting static dataset. We compare to CbAS (Brookes et al., 2019) and other baselines, including CEM (Cross Entropy Method), RWR (Reward Weighted Regression) and a method that uses a forward model – GB (Gómez-Bombarelli et al., 2018) reported by Brookes et al. (2019). For evaluation, we report

the groundtruth score of the output of optimization (max), and the 50th-percentile groundtruth score of all the samples produced via sampling (this is without inference in the MIN case) so as to be comparable to Brookes et al. (2019). In Table 5, we show that MINs are comparable to the best performing method on this task, and produce samples with the highest score among all the methods considered.

These results suggest that MINs can perform competitively with previously proposed model-based optimization methods in the active setting, reaching comparable or better performance when compared both to Bayesian optimization methods and previously proposed methods for a higher-dimensional protein design task.

## 5 DISCUSSION

In this work, we presented a novel approach towards model-based optimization (MBO). Instead of learning a proxy forward function $f_\theta(\mathbf{x})$ from inputs $\mathbf{x}$ to scores $y$, MINs learn a stochastic inverse mapping from scores $y$ to inputs. MINs are resistent to out-of-distribution inputs and can optimize over high dimensional $\mathbf{x}$ values where valid inputs lie on a narrow manifold. By using simple and principled design decisions, such as re-weighting the data distribution, MINs can perform effective model-based optimization even from static, previously collected datasets in the data-driven setting without the need for active data collection. We also described ways to perform active data collection if needed. Our experiments showed that MINs are capable of solving MBO optimization tasks in both contextual and non-contextual settings, and are effective over highly semantic score functions such as age of the person in an image.

| Method | Max | 50%ile |
|---|---|---|
| *MIN (Ours)* | **3.42** | 3.24 |
| *MIN - R* | 3.37 | **3.28** |
| *CbAS* | 3.36 | **3.28** |
| *RWR* | $\sim 3.00$ | $\sim 2.97$ |
| *CEM-PI* | $\sim 2.92$ | $\sim 2.9$ |
| *GB*[*] | $\sim 3.25$ | $\sim 3.25$ |

Table 5: Protein design results, with maximum fluorescence and the 50[th] percentile out of 100 samples. Prior method results are from Brookes et al. (2019). MINs perform comparably to CbAS. MINs without reweighting (MIN-R) lead to more consistent sample quality (higher 50%ile score), while MINs with reweighting can produce the highest scoring sample.

Prior work has usually considered MBO in the active or "on-policy" setting, where the algorithm actively queries data as it learns. In this work, we introduced the data-driven MBO problem statement and devised a method to perform optimization in such scenarios. This is important in settings where data collection is expensive and where abundant datasets exist, for example, protein design, aircraft design and drug design. Further, MINs define a family of algorithms that show promising results on MBO problems on extremely large input spaces.

While MINs scale to high-dimensional tasks such as model-based optimization over images, and are performant in both contextual and non-contextual settings, we believe there are a number of interesting open questions for future work. The interaction between active data collection and reweighting should be investigated in more detail, and poses interesting consequences for MBO, bandits and reinforcement learning. Better and more principled inference procedures are also a direction for future work. Another avenue is to study various choices of training objectives in MIN optimization.

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

## A  PROBABILISTIC INTERPRETATION OF SECTION 3.2

In this section, we show that the inference scheme described in Equation 4, Section 3.2 emerges as a deterministic relaxation of the probabilistic inference scheme described below. We re-iterate that in Section 3.2, a singleton $x^*$ is the output of optimization, however the procedure can be motivated from the perspective of the following probabilistic inference scheme.

Let $p(\mathbf{x}|y)$ denote a stochastic inverse map, and let $p_f(y|x)$ be a probabilistic forward map. Consider the following optimization problem:

$$\arg\max_{y,\hat{p}} \ \mathbb{E}_{\mathbf{x}\sim\hat{p}(\mathbf{x}|y),\hat{y}\sim p_f(\hat{y}|\mathbf{x})}[\hat{y}]$$
$$\text{such that } \mathcal{H}(\hat{y}|\mathbf{x}) \leq \epsilon_1,$$
$$D(\hat{p}(\mathbf{x}|y), p_\theta(\mathbf{x}|y)) \leq \epsilon_2,$$

where $p_\theta(\mathbf{x}|y)$ is the probability distribution induced by the learned inverse map (in our case, this corresponds to the distribution of $f_\theta^{-1}(y,z)$ induced due to randomness in $z \sim p_0(\cdot)$), $p_f(\mathbf{x}|y)$ is the learned forward map, $\mathcal{H}$ is Shannon entropy, and $D$ is KL-divergence measure between two distributions. In Equation 4, maximization is carried out over the input $y$ to the inverse-map, and the input $z$ which is captured in $\hat{p}$ in the above optimization problem, i.e. maximization over $z$ in Equation 4 is equivalent to choosing $\hat{p}$ subject to the choice of singleton/ Dirac-delta $\hat{p}$. The Lagrangian is given by:

$$\mathcal{L}(y,\hat{p};p,p_f) = \mathbb{E}_{\mathbf{x}\sim\hat{p}(\mathbf{x}|y),\hat{y}\sim p_f(\hat{y}|\mathbf{x})}[\hat{y}] + \lambda_1\left(\mathbb{E}_{\mathbf{x}\sim\hat{p}(\mathbf{x}|y),\hat{y}\sim p_f(\hat{y}|\mathbf{x})}[\log p_f(\hat{y}|\mathbf{x})] + \epsilon_1\right) +$$
$$\lambda_2\left(\epsilon_2 - D(\hat{p}(\mathbf{x}|y), p_\theta(\mathbf{x}|y))\right)$$

In order to derive Equation 4, we restrict $\hat{p}$ to the Dirac-delta distribution generated by querying the learned inverse map $f_\theta^{-1}$ at a specific value of $z$. Now note that the first term in the Lagrangian corresponds to maximizing the "reconstructed" $\hat{y}$ similarly to the first term in Equation 4. If $p_f$ is assumed to be a Gaussian random variable with a fixed variance, then $\log p_f(\hat{y}|\mathbf{x}) = -||\hat{y} - \mu(\mathbf{x})||_2^2$, where $\mu$ is the mean of the probabilistic forward map. With deterministic forward maps, we make the assumption that $\mu(\mathbf{x}) = y$ (the queried value of $y$), which gives us the second term from Equation 4.

Finally, in order to obtain the $\log p_0(z)$ term, note that, $D(\hat{p}(\mathbf{x}|y), p_\theta(\mathbf{x}|y)) \leq D(\delta_z(\cdot), p_0(\cdot)) = -\log p_0(z)$ (by the data processing inequality for KL-divergence). Hence, constraining $\log p_0(z)$ instead of the true divergence gives us a lower bound on $\mathcal{L}$. Maximizing this lower bound (which is the same as Equation 4) hence also maximizes the true Lagrangian $\mathcal{L}$.

## B  BIAS-VARIANCE TRADEOFF DURING MIN TRAINING

In this section, we provide details on the bias-variance tradeoff that arises in MIN training. Our analysis is primarily based on analysing the bias and variance in the $\ell_2$ norm of the gradient in two cases – if we had access to infinte samples of the distribution over optimal $y$s, $p^*(y)$ (this is a Dirac-delta distribution when function $f(\mathbf{x})$ evaluations are deterministic, and a distribution with non-zero variance when the function evaluations are stochastic or are corrupted by noise). Let $\hat{\mathcal{L}}_p(\mathcal{D}) = \frac{1}{|\mathcal{Y}|}\sum_{y_j\sim p_\mathcal{D}(y)} \frac{p(y_j)}{p_\mathcal{D}(y_j)}\left(\frac{1}{|N_{y_j}|}\sum_{k=1}^{|N_{y_j}|}\hat{D}(\mathbf{x}_{j,k}, f^{-1}(y_j))\right)$ denote the empirical objective that the inverse map is trained with. We first analyze the variance of the gradient estimator in Lemma B.2. In order to analyse this, we will need the expression for variance of the importance sampling estimator, which is captured in the following Lemma.

**Lemma B.1** (Variance of IS (Metelli et al., 2018)). *Let $P$ and $Q$ be two probability measures on the space $(\mathcal{X}, \mathcal{F})$ such that $d_2(P||Q) < \infty$. Let $\mathbf{x}_1, \cdots, \mathbf{x}_N$ be $N$ randomly drawn samples from $Q$, and $f : \mathcal{X} \to \mathbb{R}$ is a uniformly-bounded function. Then for any $\delta \in (0, 1]$, with probability atleast $1 - \delta$,*

$$\mathbb{E}_{x\sim P}[f(x)] \in \left[\frac{1}{N}\sum_{i=1}^{N}w_{P/Q}(x_i)f(x_i) \pm ||f||_\infty\sqrt{\frac{(1-\delta)d_2(P||Q)}{\delta N}}\right]$$

Equipped with Lemma B.1, we are ready to show the variance in the gradient due to reweighting to a distribution for which only a few datapoints are observed.

**Lemma B.2** (Gradient Variance Bound for MINs). *Let the inverse map be given by $f_\theta^{-1}$. Let $N_y$ denote the number of datapoints observed in $\mathcal{D}$ with score equal to $y$, and let $\hat{\mathcal{L}}_p(\mathcal{D})$ be as defined above. Let $\mathcal{L}_p(p_\mathcal{D}) = \mathbb{E}[\hat{\mathcal{L}}_p(\mathcal{D})]$, where the expectation is computed with respect to the dataset $\mathcal{D}$. Assume that $||\nabla_\theta \hat{D}(\mathbf{x}, f^{-1}(y))||_2 \leq L$ and $\mathrm{var}[\nabla_\theta \hat{D}(\mathbf{x}, f^{-1}(y))] \leq \sigma^2$. Then, there exist some constants $C_1, C_2$ such that with a confidence at least $1 - \delta$,*

$$\mathbb{E}\left[||\nabla_\theta \hat{\mathcal{L}}_p(\mathcal{D}) - \nabla_\theta \mathcal{L}_p(p_\mathcal{D})||_2^2\right] \quad \leq \quad C_1 \mathbb{E}_{y \sim p(y)}\left[\sigma^2 \frac{\log \frac{1}{\delta}}{N_y}\right] \quad + \quad C_2 L^2 \frac{(1-\delta) d_2(p||p_\mathcal{D})}{\delta \sum_{y \in \mathcal{D}} N_y}$$

*Proof.* We first bound the range in which the random variable $\nabla_\theta \hat{\mathcal{L}}_p(\mathcal{D})$ can take values as a function of number of samples observed for each $y$. All the steps follow with high probability, i.e. with probability greater than $1 - \delta$,

$$\nabla_\theta \hat{\mathcal{L}}_p(\mathcal{D}) = \nabla_\theta \frac{1}{|\mathcal{Y}_\mathcal{D}|} \sum_{y_j \sim p_\mathcal{D}(y)} \frac{p(y_j)}{p_\mathcal{D}(y_j)} \left(\frac{1}{|N_{y_j}|} \sum_{k=1}^{|N_{y_j}|} \hat{D}(\mathbf{x}_{j,k}, f^{-1}(y_j))\right)$$

$$\in \frac{1}{|\mathcal{Y}_\mathcal{D}|} \sum_{y_j \sim p_\mathcal{D}(y)} \left[\mathbb{E}_{\mathbf{x}_{ij} \sim p(\mathbf{x}|y_j)}\left[\hat{D}(\mathbf{x}_{ij}, y_j)\right] \pm \sqrt{\frac{\mathrm{var}(\hat{D}(x, y)) \cdot (\log \frac{!}{\delta})}{\delta \cdot N_y}}\right]$$

$$\in \mathbb{E}_{y_j \sim p(y)}\left[\mathbb{E}_{\mathbf{x}_{ij} \sim p(\mathbf{x}|y_j)}\left[\hat{D}(\mathbf{x}_{ij}, y_j)\right] \pm \sqrt{\frac{\mathrm{var}(\hat{D}(x, y)) \cdot (\log \frac{!}{\delta})}{\delta \cdot N_y}}\right] \pm \sqrt{\frac{(1-\delta) \cdot d_2(p(y)||p_\mathcal{D}(y))}{\delta \cdot \sum_{y_j \in \mathcal{D}} N_{y_j}}}$$

(5)

where $d_2(p||q)$ is the exponentiated Renyi-divergence between the two distributions $p$ and $q$, i.e. $d_2(p(y)||q(y)) = \int_y q(y) \left(\frac{p(y)}{q(y)}\right)^2 dy$. The first step follows by applying Hoeffding's inequality on each inner term in the sum corresponding to $y_j$ and then bounding the variance due to importance sampling $y$s finally using concentration bounds on variance of importance sampling using Lemma B.1.

Thus, the gradient can fluctuate in the entire range of values as defined above with high probability. Thus, with high probability, atleast $1 - \delta$,

$$\mathbb{E}\left[||\nabla_\theta \hat{\mathcal{L}}_p(\mathcal{D}) - \nabla_\theta \mathcal{L}_p(p_\mathcal{D})||_2^2\right] \leq C_1 \mathbb{E}_{y \sim p(y)}\left[\sigma^2 \frac{\log \frac{1}{\delta}}{N_y}\right] + C_2 L^2 \frac{(1-\delta) d_2(p||p_\mathcal{D})}{\delta \sum_{\mathcal{Y}_\mathcal{D}} N_y} \quad (6)$$

□

The next step is to bound the bias in the gradient that arises due to training on a different distribution than the distribution of optimal $y$s, $p^*(y)$. This can be written as follows:

$$||\mathbb{E}_{y \sim p^*(y)}[\mathbb{E}_{\mathbf{x} \sim p(\mathbf{x}|y)}[D(\mathbf{x}, y)]] - \mathbb{E}_{y \sim p(y)}[\mathbb{E}_{\mathbf{x} \sim p(\mathbf{x}|y)}[D(\mathbf{x}, y)]]||_2^2 \leq \mathrm{D}_{\mathrm{TV}}(p, p^*)^2 \cdot L. \quad (7)$$

where $\mathrm{D}_{\mathrm{TV}}$ is the total variation divergence between two distributions $p$ and $p^*$, and L is a constant that depends on the maximum magnitude of the divergence measure $D$. Combining Lemma B.2 and the above result, we prove Theorem 3.1.

## C   ARGUMENT FOR ACTIVE DATA COLLECTION VIA RANDOMIZED LABELING

In this section, we explain in more detail the randomized labeling algorithm described in Section 3.4. We first revisit Thompson sampling, then provide arguments for how our randomized labeling algorithm relates to it, highlight the differences, and then prove a regret bound for this scheme under mild assumptions for this algorithm. Our proof follows commonly available proof strategies for Thompson sampling.

---

**Algorithm 3** Thompson Sampling (TS)

---
1: Initialize a policy $\pi_a : \mathcal{X} \to \mathbb{R}$, data so-far $\mathcal{D}_0 = \{\}$, a prior over $\theta$ in $f_\theta - P(\theta^* | \mathcal{D}_0)$
2: **for** step $t$ in $\{0, \ldots, \text{T-1}\}$ **do**
3:    $\theta_t \sim P(\theta^* | \mathcal{F}_t)$   (Sample $\theta_t$ from the posterior)
4:    Query $\mathbf{x}_t = \arg\max_{\mathbf{x}} \mathbb{E}[f_{\theta_t}(\mathbf{x}) \mid \theta^* = \theta_t]$ (Query based on the posterior probability $\mathbf{x}_t$ is optimal)
5:    Observe outcome: $(\mathbf{x}_t, f(\mathbf{x}_t))$
6:    $\mathcal{D}_{t+1} = \mathcal{D}_t \cup (\mathbf{x}_t, f(\mathbf{x}_t))$
7: **end for**

---

**Notation** The TS algorithm queries the true function $f$ at locations $(\mathbf{x}_t)_{t \in \mathbb{N}}$ and observes true function values at these points $f(\mathbf{x}_t)$. The true function $f(\mathbf{x})$ is one of many possible functions that can be defined over the space $\mathbb{R}^{|\mathcal{X}|}$. Instead of representing the true objective function as a point object, it is common to represent a distribution $p^*$ over the true function $f$. This is justified because, often, multiple parameter assignments $\theta$, can give us the same overall function. We parameterize $f$ by a set of parameters $\theta^*$.

The $T$ period regret over queries $\mathbf{x}_1, \cdots, \mathbf{x}_T$ is given by the random variable

$$\text{Regret}(T) := \sum_{t=0}^{T-1} [f(\mathbf{x}^\star) - f(\mathbf{x}_t)]$$

Since selection of $\mathbf{x}_t$ can be a stochastic, we analyse **Bayes risk** (Russo & Van Roy, 2016; Russo et al., 2018), we define the Bayes risk as the expected regret over randomness in choosing $\mathbf{x}_t$, observing $f(\mathbf{x}_t)$, and over the prior distribution $P(\theta^*)$. This definition is consistent with Russo & Van Roy (2016).

$$\mathbb{E}[\text{Regret}(T)] = \mathbb{E}\left[\sum_{t=0}^{T-1} [f(\mathbf{x}^\star) - f(\mathbf{x}_t)]\right]$$

Let $\pi^{\text{TS}}$ be the policy with which Thompson sampling queries new datapoints. We do not make any assumptions on the stochasticity of $\pi^{\text{TS}}$, therefore, it can be a stochastic policy in general. However, we make 2 assumptions (A1, A2). The same assumptions have been made in Russo & Van Roy (2016).

**A1:** $\sup_{\mathbf{x}} f(\mathbf{x}) - \inf_{\mathbf{x}} f(\mathbf{x}) \leq 1$ (Difference between max and min scores is bounded by 1) – If this is not true, we can scale the function values so that this becomes true.

**A2:** Effective size of $\mathcal{X}$ is finite. [1]

TS (Alg 3) queries the function value at $\mathbf{x}$ based on the posterior probability that $\mathbf{x}$ is optimal. More formally, the distribution that TS queries $\mathbf{x}_t$ from can be written as: $\pi_t^{\text{TS}} = \mathbb{P}(\mathbf{x}^* = \cdot | \mathcal{D}_t)$. When we use parameters $\theta$ to represent the function parameter, and thus this reduces to sampling an input that is optimal with respect to the current posterior at each iteration: $\mathbf{x}_t \in \arg\max_{\mathbf{x} \in \mathcal{X}} \mathbb{E}[f_{\theta_t}(\mathbf{x}) | \theta^* = \hat{\theta}_t]$.

MINs (Alg 2) train inverse maps $f_\theta^{-1}(\cdot)$, parameterized as $f_\theta^{-1}(z, y)$, where $y \in \mathbb{R}$. We call an inverse map *optimal* if it is uniformly optimal given $\theta_t$, i.e. $||f_{\theta_t}^{-1}(\max_{\mathbf{x}} f(\mathbf{x}) | \theta_t) - \delta\{\arg\max_{\mathbf{x}} \mathbb{E}[f(\mathbf{x}) | \theta_t]\}|| \leq \varepsilon_t$, where $\varepsilon_t$ is controllable (usually the case in supervised learning, errors can be controlled by cross-validation).

Now, we are ready to show that the regret incurred the randomized labelling active data collection scheme is bounded by $\mathcal{O}(\sqrt{T})$. Our proof follows the analysis of Thompson sampling presented in Russo & Van Roy (2016). We first define *information ratio* and then use it to prove the regret bound.

**Information Ratio** Russo & Van Roy (2016) related the expected regret of TS to its expected information gain i.e. the expected reduction in the entropy of the posterior distribution of $\mathcal{X}^*$.

---

[1] By effective size we refer to the intrinsic dimensionality of $\mathcal{X}$. This doesn't necessarily imply that $\mathcal{X}$ should be discrete. For example, under linear approximation to the score function $f_\theta(\mathbf{x})$, i.e., if $f_\theta(\mathbf{x}) = \theta^T \mathbf{x}$, this defines a polyhedron but just analyzing a finite set of just extremal points of the polyhedron works out, thus making $|\mathcal{X}|$ effectively finite.

Information ratio captures this quantity, and is defined as:

$$\Gamma_t := \frac{\mathbb{E}_t \left[ f(\mathbf{x}_t) - f(\mathbf{x}^\star) \right]^2}{I_t \left( \mathbf{x}^\ast; (\mathbf{x}_t, f(\mathbf{x}_t)) \right)}$$

where $I(\cdot, \cdot)$ is the mutual information between two random variables and all expectations $\mathbb{E}_t$ are defined to be conditioned on $\mathcal{D}_t$. If the information ratio is small, Thompson sampling can only incur large regret when it is expected to gain a lot of information about which $\mathbf{x}$ is optimal. Russo & Van Roy (2016) then bounded the expected regret in terms of the maximum amount of information any algorithm could expect to acquire, which they observed is at most the entropy of the prior distribution of the optimal $\mathbf{x}$.

**Lemma C.1 (Bayes-regret of vanilla TS)**(Russo & Van Roy, 2016)). *For any $T \in \mathbb{N}$, if $\Gamma_t \leq \overline{\Gamma}$ (i.e. information ratio is bounded above) a.s. for each $t \in \{1, \dots, T\}$,*

$$\mathbb{E}[Regret(T, \pi^{\mathrm{TS}})] \leq \sqrt{\overline{\Gamma} H \left( \mathcal{X}^\ast \right) T}$$

We refer the readers to the proof of Proposition 1 in Russo & Van Roy (2016). The proof presented in Russo & Van Roy (2016) does not rely specifically on the property that the query made by the Thompson sampling algorithm at each iteration $\mathbf{x}_t$ is posterior optimal, but rather it suffices to have a bound on the maximum value of the information ratio $\Gamma_t$ at each iteration $t$. Thus, if an algorithm chooses to query the true function at a datapoint $\mathbf{x}_t$ such that these queries always contribute in learning more about the optimal function, i.e. $I(\cdot, \cdot)$ appearing in the denominator of $\Gamma$ is always more than a threshold, then information ratio is lower bounded, and that active data collection algorithm will have a sublinear asymptotic regret. We are interested in the case when the active data collection algorithm queries a datapoint $\mathbf{x}_t$ at iteration $t$, such that $\mathbf{x}_t$ is the optimum for a function $\hat{f}_{\hat{\theta}_t}$, where $\hat{\theta}_t$ is a sample from the posterior distribution over $\theta_t$, i.e. $\hat{\theta}_t$ lies in the high confidence region of the posterior distribution over $\theta_t$ given the data $\mathcal{D}_t$ seen so far. In this case, the mutual information between the optimal datapoint $\mathbf{x}^\star$ and the observed $(\mathbf{x}_t, f(\mathbf{x}_t))$ input-score pair is likely to be greater than 0. More formally,

$$I_t(\mathbf{x}^\star, (\mathbf{x}_t, f(\mathbf{x}_t))) \geq 0 \quad \forall \; \mathbf{x}_t = \arg \max_{\mathbf{x}} f_{\hat{\theta}_t}(\mathbf{x}) \; \text{where} \; P(\hat{\theta}_t | \mathcal{D}_t) \geq \epsilon_{\mathrm{threshold}} \qquad (8)$$

The randomized labeling scheme for active data collection in MINs performs this step. The algorithm samples a bunch of $(\mathbf{x}, y)$ datapoints, sythetically generated, – for example, in our experiments, we add noise to the values of $\mathbf{x}$, and randomly pair them with unobserved or rarely observed values of $y$. If the underlying true function $f$ is smooth, then there exist a finite number of points that are sufficient to uniquely describe this function $f$. One measure to formally characterize this finite number of points that are needed to uniquely identify all functions in a function class is given by *Eluder dimension* (Russo & Van Roy).

By augmenting synthetic datapoints and training the inverse map on this data, the MIN algorithm ensures that the inverse map is implicitly trained to be an accurate inverse for the unique function $f_{\hat{\theta}_t}$ that is consistent with the set of points in the dataset $\mathcal{D}_t$ and the augmented set $\mathcal{S}_t$. Which sets of functions can this scheme represent? The functions should be consistent with the data seen so far $\mathcal{D}_t$, and can take randomly distributed values outside of the seen datapoints. This can roughly argued to be a sample from the posterior over functions, which Thompson sampling would have maintained given identical history $\mathcal{D}_t$.

**Lemma C.2 (Bounded-error training of the posterior-optimal $\mathbf{x}_t$ preserves asymptotic Bayes-regret).** *$\forall t \in \mathbb{N}$, let $\hat{\mathbf{x}}_t$ be any input such that $f(\hat{\mathbf{x}}_t) \geq \max_{\mathbf{x}} \mathbb{E}[f(\mathbf{x})|\mathcal{D}_t] - \varepsilon_t$. If MIN chooses to query the true function at $\hat{\mathbf{x}}_t$ and if the sequence $(\varepsilon_t)_{t \in \mathbb{N}}$ satisfies $\sum_{t=0}^{T} \varepsilon_t = \mathcal{O}(\sqrt{T})$, then, the regret from querying this $\varepsilon_t$-optimal $\hat{\mathbf{x}}_t$ which is denoted in general as the policy $\hat{\pi}^{\mathrm{TS}}$ is given by $\mathbb{E}[Regret(T, \hat{\pi}^{\mathrm{TS}})] = \mathcal{O}(\sqrt{T})$.*

*Proof.* This lemma intuitively shows that if posterior-optimal inputs $\mathbf{x}_t$ can be "approximately" queried at each iteration, we can still maintain sublinear regret. To see this, note:

$$f(\mathbf{x}^\star)) - f(\hat{\mathbf{x}}_t) = f(\mathbf{x}^\star) - f(\mathbf{x}_t) + f(\mathbf{x}_t) - f(\hat{\mathbf{x}}_t).$$

$$\implies \mathbb{E}[\text{Regret}(T, \hat{\pi}^{\text{TS}})] = \mathbb{E}[\text{Regret}(T, \pi^{\text{TS}})] + \mathbb{E}[\sum_{t=1}^{T}(f(\mathbf{x}_t) - f(\hat{\mathbf{x}}_t))]$$

The second term can be bounded by the absolute value in the worst case, which amounts $\sum_{t=0}^{T} \varepsilon_t$ extra Bayesian regret. As Bayesian regret of TS is $\mathcal{O}(\sqrt{T})$ and $\sum_{t=0}^{T} \varepsilon_t = \mathcal{O}(\sqrt{T})$, the new overall regret is also $\mathcal{O}(\sqrt{T})$. □

**Theorem C.3 (Bayesian Regret of randomized labeling active data collection scheme proposed in Section 3.4 is $\mathcal{O}(\sqrt{T})$).** *Regret incurred by the MIN algorithm with randomized labeling is of the order $\mathcal{O}(\sqrt{(\bar{\Gamma} H(\mathcal{X}^*) + C)T})$.*

*Proof.* Simply put, we will combine the insight about the mutual information $I(\mathbf{x}^{\star}, (\mathbf{x}_t, f(\mathbf{x}_t))) > 0$ and C.2 in this proof. Non-zero mutual information indicates that we can achieve a $\mathcal{O}(\sqrt{T})$ regret if we query $\mathbf{x}_t$s which are optimal corresponding to some implicitly defined forward function lying in the high confidence set of the true posterior given the observed datapoints $\mathcal{D}_t$. Lemma C.2 says that if bounded errors are made in fitting the inverse map, the overall regret remains $\mathcal{O}(\sqrt{T})$.

More formally, if $||f_{\theta_t}^{-1}(\max_{\mathbf{x}} f(\mathbf{x})|\theta_t) - \delta\{\arg\max_{\mathbf{x}} \mathbb{E}[f(\mathbf{x})|\theta_t]\}|| \leq \delta_t$, this means that

$$||\mathbb{E}_{\mathbf{x}_t \sim f_{\theta_t}^{-1}}[f(\mathbf{x}_t)] - \mathbb{E}_{\mathbf{x}_t' \sim \pi_t^{\text{TS}}}[f(\mathbf{x}_t')]|| \leq ||f(\cdot)||_{\infty} \cdot ||f_{\theta_t}^{-1} - \pi_t^{\text{TS}}|| \leq \delta_t R_{\max} \leq \varepsilon_t$$

and now application of Lemma C.2 gives us the extra regret incurred. (Note that this also provides us a way to choose the number of training steps for the inverse map)

Further, note if we sample $\mathbf{x}_t$ at iteration $t$ from a distribution that shares support with the true posterior over optimal $\mathbf{x}_t$ (which is used by TS), we still incur sublinear, bounded $\mathcal{O}(\sqrt{\bar{\Gamma} H(A^*)T})$ regret.

In the worst case, the overall bias caused due to the approximations will lead to an additive cumulative increase in the Bayesian regret, and hence, there is a constant $\exists \; C \geq 0$, such that $\mathbb{E}[\text{Regret}(T, f^{-1})] = \mathcal{O}(\sqrt{(\bar{\Gamma} H(\mathcal{X}^*) + C)T})$. □

# D  ADDITIONAL EXPERIMENTS AND DETAILS

## D.1  CONTEXTUAL IMAGE OPTIMIZATION

In this set of static dataset experiments, we study contextual MBO tasks on image pixels. Unlike the contextual bandits case, where $\mathbf{x}$ corresponds to an image label, here $\mathbf{x}$ corresponds to entire images. We construct several tasks. First, we study stroke width optimization on MNIST characters, where the context is the class of the digit we wish to optimize. Results are shown in Figure 4. MINs correctly produce digits of the right class, and achieve an average score over the digit classes of **237.6**, whereas the average score of the digits in the dataset is **149.0**.

The next task is to test the ability of MINs to be able to complete/inpaint unobserved patches of an image given an observed context patch. We use two masks: *mask A*: only top half and *mask B*: only top one-fourth parts of the image are visible, to mask out portions of the image and present the masked image as context $c$ to the MIN, with the goal being to produce a valid completion $\mathbf{x}$, while still maximizing score corresponding to the stroke width. We present some

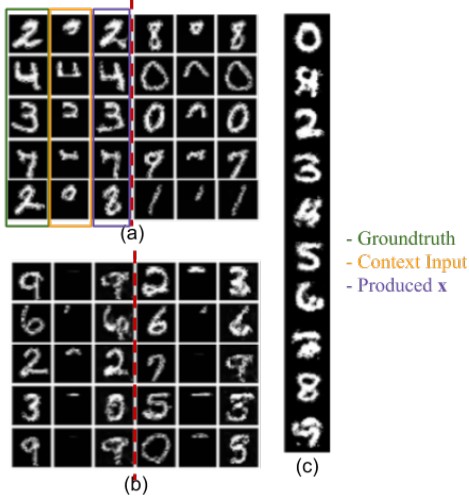

Figure 4: Contextual MBO on MNIST. In (a) and (b), top one-half and top one-fourth of the image respectively and in (c) the one-hot encoded label are provided as contexts. The goal is to produce the maximum stroke width character that is valid given the context. In (a) and (b), we show triplets of the groundtruth digit (green), the context passed as input (yellow) and the produced images $\mathbf{x}$ from the MIN model (purple).

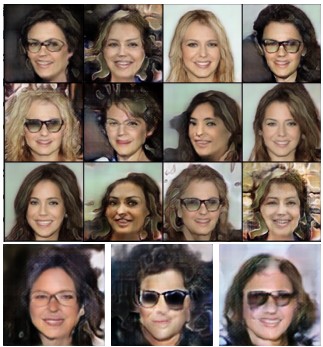

Figure 5: Additional results for non-contextual image optimization. This task is performed on the CelebA dataset. The aim is to maximize the score of an image which is given by the sum of attributes: eyeglasses, smiling, wavy hair and no beard. MINs produce optimal $\mathbf{x}$ – visually these solutions indeed optimize the score.

sample completions in Figure 4. The quantitative results are presented in Table 6. We find that MINs are effective as compared completions for the context in the dataset in terms of score while still producing a visibly valid character.

We evaluate MINs on a complex semantic optimization task on the CelebA (Liu et al., 2015) dataset. We choose a subset of attributes and provide their one-hot encoding as context to the model. The score is equal to the $\ell_1$ norm of the binary indicator vector for a different subset of attributes disjoint from the context. We present our results in Figure 3. We observe that MINs produce diverse images consistent with the context, and is also able to effectively infer the score function, and learn features to maximize it. Some of the model produced optimized solutions were presented in Section 4 in Figure 3. In this section, we present the produced generations for some other contexts. Figure 7 shows these results.

## D.2 ADDITIONAL RESULTS FOR NON-CONTEXTUAL IMAGE OPTIMIZATION

In this section, we present some additional results for non-contextual image optimization problems. We also evaluated our contextual optimization procedure on the CelebA dataset in a non-contextual setting. The reward function is the same as that in the contextual setting – the sum of attributes: wavy hair, no beard, smiling and eyeglasses. We find that MINs are able to sucessfully produce solutions in this scenario as well. We show some optimized outputs at different iterations from the model in Figure 5.

**cGAN baseline.**    We compare our MIN model to a cGAN baseline on the IMDB-Wiki faces dataset for the semantic age optimization task. In general, we found that the cGAN model learned to ignore the score value passed as input even when trained on the entire dataset (without excluding the youngest faces) and behaved almost like a regular unconditional GAN model when queried to produce images $\mathbf{x}$ corresponding to the smallest age. We suspect that this could possibly be due to the fact that age of a person doesn't have enough direct signal to guide the model to utilize it unless other tricks like reweighting proposed in Section 3.3 which explicitly enforce the model attention to datapoints of interest, are used. We present the produced optimized $\mathbf{x}$ in Figure 6.

## D.3 QUANTITATIVE SCORES FOR NON-CONTEXTUAL MNIST OPTIMIZATION

In Figure 8, we highlight the quantitative score values for the stroke width score function (defined as the number of pixels which have intensity more than a threshold). Note that MINs achieve the highest value of average score while still resembling a valid digit, that stays inside the manifold of valid digits, unlike a forward model which can get high values of the score function (number of pixels turned on), but doesn't stay on the manifold of valid digits.

## D.4 EXPERIMENTAL DETAILS AND SETUP

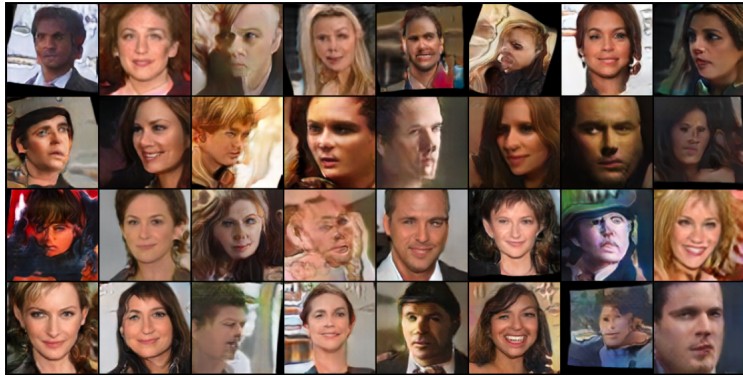

Figure 6: Optimal **x** solutions produced by a cGAN for the youngest face optimization task on the IMDB-faces dataset. We note that a cGAN learned to ignore the score value and produced images as an unconditional model, without any noticeable correlation with the score value. The samples produced mostly correspond to the most frequently occurring images in the dataset.

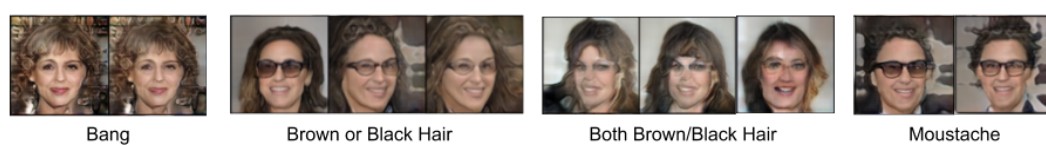

Figure 7: Images returned by the MIN optimization for optimization over images. We note that MINs perform successful optimization over the an objective defined by the sum of desired attributes. Moreover, for unseen contexts, such as both brown and black hair, the optimized solutions look aligning with the context reasonably, and optimize for the score as well.

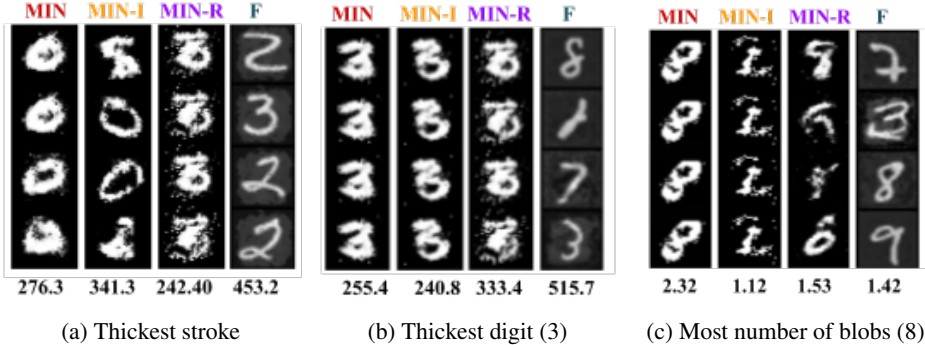

| (a) Thickest stroke | (b) Thickest digit (3) | (c) Most number of blobs (8) |
| --- | --- | --- |

Figure 8: Results for non-contextual static dataset optimization on MNIST annotated with quantitative score values achieved mentioned below each figure.

Table 6: Average quantitative performance for MNIST inpainting

| Mask | MIN | Dataset |
| --- | --- | --- |
| *mask A* | **223.57** | 149.0 |
| *mask B* | **234.32** | 149.0 |

In this section, we explain the experimental details and the setup of our model. For our experiments involving MNIST and optimization of benchmark functions task, we used the same architecture as a fully connected GAN - where the generator and discriminator are both fully connected networks. We based our code for this part on the open-source implementation (Linder-Norén). For the forward model experiments in these settings, we used a 3-layer feedforward ReLU network with hidden units of size 256 each in this setting. For all experiments on CelebA and IMDB-Wiki faces, we used the VGAN (Peng et al., 2019) model and the associated codebase as our starting setup. For experiments on batch contextual bandits, we used a fully connected discriminator and generator for MNIST, and a convolutional generator and Resnet18-like discriminator for CIFAR-10. The prediction in this setting is categorical – 1 of 10 labels needs to be predicted, so instead of using reinforce or derivative free optimization to train the inverse map, we used the Gumbel-softmax Jang et al. (2016) trick with a temperature $\tau = 0.75$, to be able to use stochastic gradient descent to train the model. For the protein flourescence maximization experiment,

we used a 2-layer, 256-unit feed-forward gumbel-softmax inverse map and a 2-layer feed-forward discriminator.

We trained models present in open-source implementations of BanditNet (Sachdeva), but were unable to reproduce results as reported by Joachims et al. (2018). Thus we reported the paper reported numbers from the BanditNet paper in the main text as well.

Temperature hyperparameter $\tau$ which is used to compute the reweighting distribution is adaptively chosen based on the $90^{\text{th}}$ percentile score in the dataset. For example, if the difference between $y_{max}$ and $y_{90^{\text{th}}-\text{percentile}}$ is given by $\alpha$, we choose $\tau = \alpha$. This scheme can adaptively change temperatures in the active setting. In order to select the constant which decides whether the bin corresponding to a particular value of $y$ is small or not, we first convert the expression $\frac{N_y}{N_y + \lambda}$ to use densities rather than absolute counts, that is, $\frac{\hat{p}_{\mathcal{D}}(y)}{\hat{p}_{\mathcal{D}}(y) + \lambda}$, where $\hat{p}_{\mathcal{D}}(y)$ is the empirical density of observing $y$ in $\mathcal{D}$, and now we use the same constant $\lambda = 0.003$. We did not observe a lot of sensitivity to $\lambda$ values in the range $[0.0001, 0.007]$, all of which performed reasonably similar. We usually fixed the number of bins to 20 for the purposed of reweighting, however note that the inverse map was still trained on continuous $y$ values, which helps it extrapolate.

In the active setting, we train two copies of $f^{-1}$ jointly side by side. One of them is trained on the augmented datapoints generated out of the randomized labelling procedure, and the other copy is just trained on the real datapoints. This was done so as to prevent instabilities while training inverse maps. Training can also be made more incremental in this manner, and we need to train an inverse map to optimality inside every iteration of the active MIN algorithm, but rather we can train both the inverse maps for a fixed number of gradient steps.

