# OpenReview forum: "Model Inversion Networks for Model-Based Optimization"
_ICLR.cc/2020/Conference — Reject_

### Official Review · AnonReviewer3 · 2019-10-22
**Official Blind Review #3**

**Rating:** 6

**Review:**

In this paper, the authors present a method for model based blackbox optimization for cases where the design variables x are likely to lie on a manifold within a high dimensional space. The basic idea is to use a generative model to map targets y to potential design variables that would produce that target, and use this generative model in combination with a trained forward model to produce an optimal setting x* that both lies on the manifold and takes a high y* value, possibly higher than seen in the collected dataset (see the first two terms in equation 4).

Overall, I feel positively about the paper, largely because of the idea it introduces. In particular, the task of manifold constrained optimization is not highly studied in BayesOpt, and it probably should be. Specifically, BayesOpt *is* occasionally studied on problems where the inputs are natural images, protein sequences, controller policies, and other settings where not all inputs are going to be valid, even if all inputs can technically give rise to a function value. The use of a GAN to constrain the possible inputs via a learned measure p(x|y) is a nice idea where applicable, and represents a concrete utility for GANs beyond generating images. All the individual components the authors' use to put this together are somewhat basic; however, the experimental results are vaguely convincing enough and I would choose to evaluate the merits of this paper on the core idea presented rather than whether or not the latest GAN architecture was used in the actual implementation.

With that said, I have a few comments. First, the constructions in 3.2 and 3.3 are perhaps a little more ad hoc than they need to be. If we are free to use probabilistic machinery for the forward model, then the fact that the inverse model effectively gives us a distribution fit to p(x|y) suggests that this can be naturally incorporated in to existing BayesOpt schemes. It might seem almost more satisfying to see a simple existing bayesopt pipeline extended with this idea, rather than wholly inference/recommendation and acquisition schemes introduced.

Additionally, two of the sections in the experimental results seemed somewhat rushed and need significant additional detail. The setup in the first portion of 4.1 is largely left to the reader to read Joachims et al., 2018 -- I'd like to see a more self contained description of the task. The more egregious example of this is the "protein floresence [sic] maximization" section. One of the baseline methods (GB) exists only as an acronym in Table 3 with no citation or discussion, and in general the section could be significantly expanded.

This last paragraph is something of a shame and one of the main weaknesses I see with the paper, as these two section are among the only quantitative results for the authors' optimization algorithm. Results are reported on Branin and Hartmann6, but frankly I don't see the value of these because they are far outside the intended application domain for the authors' work, as they are functions that are specifically intended to be optimized over compact domains. In general, it would be better to see more emphasis placed on quantitative experiments, particularly when the task is optimization. Perhaps adversarial image generation (where x arguably leaves the manifold of natural images, but not so far as to be random noise) or some other image task might substitute for the benchmark functions?



**Experience Assessment:**

I have published in this field for several years.

**Review Assessment: Checking Correctness Of Derivations And Theory:**

I assessed the sensibility of the derivations and theory.

**Review Assessment: Checking Correctness Of Experiments:**

I carefully checked the experiments.

**Review Assessment: Thoroughness In Paper Reading:**

I read the paper thoroughly.

---

> ### Author Response · Authors · 2019-11-14
> **Response to Review #3**
>
> Thank you for the valuable feedback. We have included quantitative results for our experiments in Section 4 (Table 1, 2, 3, 5 and Figure 1). We have clarified details about tasks in the experimental section. We have also added a probabilistic interpretation for the method described in Section 3.2 in Appendix A.  We answer specific questions below:
>
> Q: Constructions in 3.2 and 3.3 seem a little more adhoc than what they should be.
> A: Thank you for your feedback. While we agree there are a considerable number of details, described in Section 3.2 and 3.3, that are needed to make our method work, we believe that these details are in fact necessary for the method to be effective and we do not see how to easily retrofit them into an existing BayesOpt method. We have sought to revise the paper to avoid the impression that these details are ad hoc by adding a probabilistic interpretation of Section 3.2 in Appendix A.
>
> - In Section 3.2, we proposed a scheme for choosing the values of $y$ with which to query the inverse map. To address your comment, we have added a probabilistic interpretation of the inference scheme described in Section 3.2 in the newly added Appendix A, which is more general and perhaps more interpretable.
>
> - In Section 3.3, we described a scheme for reweighting the training set for the inverse map. Reweighting is necessary because the inverse-map, in principle, could be a multimodal and non-smooth function. Modeling the entire inverse map is unnecessary, as our goal is optimization and not inverse design. To justify the specific design choice of our reweighting scheme, we motivated it from the perspective of the bias-variance tradeoff in the training objective or its gradients. As described in Section 3.3, training under the distribution centered around or at the singleton value, $y^*$ leads to a high variance training objective. This phenomenon is agnostic to the particular choice of the objective.
>
> Hence, we argue that these choices have a theoretical justification. In our experiments, the effects of reweighting and inference are evident across a diverse range of contextual/non-contextual tasks:
> (1) batch contextual bandits (Table 1: MIN, MIN w/o inference, MIN w/o reweighting)
> (2) character stroke width optimization (Figure 1: MIN, MIN w/o inference, MIN w/o reweighting)
> (3) Semantic score optimization task (Figure 2 vs Figure 6 and Table 2)
> (4) Protein fluorescence maximization task (Table 5)
>
> Q. Incorporating into existing Bayes Opt pipeline.
> A: While this point is appreciated, we were not sure how an inverse map would fit into an existing BayesOpt pipeline. One approach could be using p(x|y) to give us p(y|x) via Bayes rule, but the integral in the denominator is still intractable. We would be interested in hearing your thoughts about integrating the inverse model into existing BayesOpt schemes -- we agree this could be a promising approach, but we do not know how to realize it right now.
>
> Q: Clarifications in the experimental section.
> A: We have updated the experimental section with the following changes:
> (1) Self-contained description of Joachims et.al. contextual bandit task
> (2) Added description of the missing reference in Protein Fluorescence maximization task; described the model architecture; improved the task description.
>
> Q: Quantitative results for optimization tasks
> A: We have revised the paper to add quantitative results for all of the experiments, such that there are now five separate experiments with quantitative results. We hope that this addresses your concern about the quantitative evaluation, and we would welcome any further suggestions. These optimization scores were reported in the appendix in the submitted version. But now we have included these in the main paper (Table 2 and Table 3, Figure 1). A list of all quantitative results in the revised version of the paper is as follows --
>
> (1) Table 1 (Batch Contextual Bandits): We report the average test accuracy on unseen, novel contexts from the test split. The score is the average  0-1 reward across the test split, which is also the average test accuracy.
> (2) Figure 1: We have added scores beneath each method. Higher scores are better, subject to the produced images being valid digit images.
> (3) Table 2: We have added scores for the youngest face optimization task, where the score is given by the negative age of the face in the picture.
> (4) Table 3: We have added scores for the contextual MNIST inpainting for two sub-cases.
> (5) Table 5: Quantitative scores for protein design task.
>
> In all, there are five experiments with quantitative results.

---

### Official Review · AnonReviewer2 · 2019-10-24
**Official Blind Review #2**

**Rating:** 3

**Review:**

This paper tackles the problem of solving a black-box optimization problem where only some samples have been observed. This task requires a good model that can be both expressive and generalizable. Instead of learning only a single forward model of x -> y, this paper proposes to additionally use a mapping from y -> x. Optimizing in the space of z instead of x can be much simpler, and this should also act as a strong regularizer during training. Specifically, the paper uses a GAN that transforms [y,z] -> x, where z is stochastically sampled. This paper further proposes a reweighting scheme that interpolates between a uniform weighting and weighting the best sample so far, as well as a sampling procedure that iteratively samples points and refits a second model, which was inspired by Thompson sampling.

Pros:
 - The proposed idea of using an inverse mapping is straightward but shown to be effective. The methods to make this work, namely reweighting and the randomized labeling procedure, seem to have some amount of theory behind them, though their presentation was confusing without multiple read-throughs.
 - There are plenty of experiments across a wide array of domains, including images, 2D and 6D functions, and proteins which have a discrete representation.
 - There are some comparisons to Spearmint and a scalable BO method (DNGO), though only on the 2D and 6D functions.

Cons:
 - The proposed pieces were often difficult to follow, and there doesn't seem to be sufficient information regarding the reweighting and randomized labeling for understanding and reproducing this work (see Questions).
 - Only a few ablation experiments were carried out, and the effect of reweighting seems to only appear as a visual comparison in Figure 1. As the method seems quite different from related approaches, having more systematic comparisons would give us a better understanding of the core sources of improvement and better instigate future works.

Questions:
- After Theorem 3.1, what is g? A linear function can only change a pdf by its bias (as any multiplication only affects the normalization constant), so not sure how to interpret g. I wasn't able to find this in the Appendix either.
- When creating the augmented dataset for defining & approximating some kind of posterior, can you be more concrete about how the y (and x) values are chosen? More specifically, is this procedure approximating some kind of meaningful posterior? If the y values are sampled randomly (from some fixed prior or perhaps from the p(y) described in the reweighting section?), what makes this procedure meaningful (does it rely on the learned GAN somehow)? Seeing a simple example in 1D would be a good visualization.
- What are the different rows of Figure 1 (different initializations)? Why are the different rows so similar for MIN but not for F? Also, what were the original examples (are they closer to the F results, or the MIN results)?
- For the protein task, how did you structure the output of f^-1(x) and how did you backpropagate through the discrete variables?
- In the experiments, it wasn't explicitly clear what the "MIN without inference" setting referred to; I assume this is where a single sample from the inverse mapping GAN was used?

Additional Comments:
- When discussing the reweighted objective, the notation is a little confusing. The variables (j,k) is a different parameterization of the index i, but initially I thought k was indexing features of x. Perhaps a brief explanation of the rearrangement would make this clearer at first glance, or even remove this from the main text? (Without going through the proof, it isn't clear why this rearrangement is discussed.)
- After (1), it wasn't yet clear why this is called "model-based" optimization, as no model has been introduced yet.
- Are BO methods truly not applicable to the static setting? The static setting seems to be just a single prediction, which seems like a single step (ie. special case) of the non-static case?
- typo: "method to solve perform optimization"
- typo: Figure 1 "Obsere"

**Experience Assessment:**

I have read many papers in this area.

**Review Assessment: Checking Correctness Of Derivations And Theory:**

I assessed the sensibility of the derivations and theory.

**Review Assessment: Checking Correctness Of Experiments:**

I assessed the sensibility of the experiments.

**Review Assessment: Thoroughness In Paper Reading:**

I read the paper at least twice and used my best judgement in assessing the paper.

---

> ### Author Response · Authors · 2019-11-07
> **Response to Review #2**
>
> Thank you for the constructive feedback. We have updated the paper (changes in red) to address the clarity concerns. Concisely, we have addressed clarity issues (in red) along these directions:
> 1. Added interpretation of the function $g$ in reweighting (Section 3.3)
> 2. Described the procedure for creating the augmented dataset in practice (Section 3.5)
> 3. Added clear description to Figure 1 to describe how optimization is performed with MINs w/o inference (MIN - I); added a description of different rows in the caption of Figure 1.
> 4. Improved Section 3.3 by omitting the rearrangement of the loss term
> 5. Added description of the architecture of $f^{-1}$ in Section 4.2 (protein fluorescence maximization). This has already been described in Appendix C.4.
>
> Regarding the concerns on ablations, we first summarize the existing ablations in our submission.
> 1. Table 1 (we compare MINs and MINs w/o inference)
> 2. Figure 1 (we compare MIN, MIN w/o inference, MIN w/o reweighting). Figure 8 (Appendix) shows quantitative scores for this task (measured as the number of active pixels (pixel value > 0))
> 3. Figure 2 and Figure 6, Appendix C.2 (Compares MIN and cGAN (no reweighting) on the youngest face optimization task; reweighting seems essential in this case)
> 4.Table 4 (final) (Ablation of randomized labeling; MIN + greedy = no randomized labeling)
> 5. Table 5 (final) (w/o reweighting)
>
> We have updated the paper with an ablation study of MINs without reweighting on the contextual bandit task (Table 1). Also, please find gifs of runs (given different initial points) of the randomized labeling on a 1D function:  https://ibb.co/album/i4d8qa (Earlier at: https://tinyurl.com/min-randomized1 https://tinyurl.com/min-randomized2 and https://tinyurl.com/min-randomized3) (For reference, we used the same 1D function as https://tinyurl.com/min-1d-example )
>
> Please let us know if more ablations are necessary.  Please let us know if any further clarity issues remain, we are happy to further update the paper. Further, all code used to run experiments will be released with the final version of the paper.
>
> Please find specific replies to other questions below.
>
> Q: Only a few ablation experiments were carried out, and the effect of reweighting seems to only appear as a visual comparison in Figure 1.
> We have included an ablation of reweighting on the batch contextual bandit task (Table 1). MINs with reweighting outperforms MIN w/o reweighting on CIFAR 10. Figure 6, Appendix C.2 shows a comparison to a cGAN baseline without any reweighting on the youngest face optimization task (compare to Figure 2). Without reweighting, the model tends to ignore the score value, leading to very poor optimization performance.
>
> Q: Meaningful posteriors with randomized labeling
> We provide guarantees for the randomized labeling procedure in Appendix B, where we show that, as long as the newly sampled datapoint lies in the support of the posterior, our algorithm recovers Bayesian Regret guarantees similar to TS. We have also included a simple 1D example as a visualization.
>
> Q: Why is this called Model-based optimization?
> The problem setting is called model-based optimization usually because most of the approaches solving such optimization problems rely on learning a model $\hat{f}(x)$, from observed datapoint pairs $(x_i, y_i)$, and then find the optimum of this learned function. Bayesian Optimization is also a special case of this procedure. These definitions can be found in the papers referenced below. MINs learn an inverse-map instead of a forward function.
>
> [1] mlrMBO: A Modular Framework for Model-Based Optimization of Expensive Black-Box Functions, Biscl et al. https://arxiv.org/abs/1703.03373
> [2] A survey of Model-based Methods for Global Optimization, Beielstein. https://pdfs.semanticscholar.org/ae2d/95955f54d647fe28edc5fc3658622b040b69.pdf
>
> Q: What are the different rows of Figure 1 (different initializations)?
> Different rows in Figure 1 present different outputs produced from the optimization procedure. MIN produces identical outputs as the inference procedure converges to the same x (digit image). When not using the specialized inference procedure described in Section 3.2 (MIN - I), we just sample x (digit images) from the inverse map for high values of y. In this case, there are multiple x’s that could be produced, and the rows present these samples. F produces images of invalid digits.
>
> Q: Are BO methods truly not applicable to the static setting?
> To the best of our knowledge, we haven’t seen a BO method applied in the static setting. Conditional Neural Processes (Garnelo et.al.) and Neural Processes (Garnelo et.al.) can perhaps be regarded as closest, but the main aim of these papers is to learn the forward function with correct uncertainty quantification. These approaches do not aim to perform optimization over the input $x$. If the reviewer has specific papers in mind, we would be happy to include those in related work.

---

### Official Review · AnonReviewer1 · 2019-10-24
**Official Blind Review #1**

**Rating:** 1

**Review:**

The paper prposes to learn an inverse network to predict x given a target y for optimisation, instead of the traditional way of optimisation (e.g. using Bayesian optimisation for the complex cases considered in the paper). However, unfortunately, this paper is too close in concept, and in my understanding lower in the solution quality to this recent paper:

Nguyen, Phuoc, Truyen Tran, Sunil Gupta, Santu Rana, Matthew Barnett, and Svetha Venkatesh. "Incomplete conditional density estimation for fast materials discovery." In Proceedings of the 2019 SIAM International Conference on Data Mining, pp. 549-557. Society for Industrial and Applied Mathematics, 2019.

Please let me know if I missed anything. Otherwise it is a reject from me.

**Experience Assessment:**

I have published one or two papers in this area.

**Review Assessment: Checking Correctness Of Derivations And Theory:**

I assessed the sensibility of the derivations and theory.

**Review Assessment: Checking Correctness Of Experiments:**

I assessed the sensibility of the experiments.

**Review Assessment: Thoroughness In Paper Reading:**

I read the paper at least twice and used my best judgement in assessing the paper.

---

> ### Author Response · Authors · 2019-11-06
> **Response to Review #1**
>
> Thank you for this very relevant reference -- we were not aware of this prior work and will add and discuss it in our related work section. However, we believe that this work is broadly different both in terms of the problem it is addressing and the method that it is proposing.
>
> Summary of our response: Nguyen et al. does not perform model-based optimization to produce an input $x$ that maximizes a scalar-valued $y$, but rather learns a generative inverse function of the form $f: y \rightarrow x$, where $y$ is a vector-valued context, not the value of a function in an optimization problem. Further, there are many important differences in the technical approach which are elaborated upon below. Our paper already includes a comparison that is analogous to the most obvious application of this method for model-based optimization -- Figure 6 (Appendix C.2). MINs (Figure 2), which are designed for model-based optimization perform substantially better.
>
> Detailed response:
> -----------------------------------------------------------------
> We describe the differences in detail below:
>
> 1. Differences in Problem Statement: Our aim is to solve optimization problems of the form $x^* = \arg \max_x f(x)$, where $f(x)$ (also referred to as y) is a scalar score value. Our goal is to find an x that *optimizes* the value of the score function. This is in contrast with the goal of Nguyen et al., which is to learn a generative model to generate values $x$ conditioned on a context $y$, where $y$ is a vector (for example, phase-transition graphs), and the goal is to produce any $x$ that is suitable for $y$. This problem statement is broadly distinct from model-based optimization.
>
> 2. Comparison of the technical approaches: Nguyen et.al.'s approach is loosely analogous to conditional generative models (for example, cGANs [Mirza and Osindero, 2014]). Since MINs are designed to solve for the optimum $x$, there are many components in the MIN procedure that make it different from a cGAN (Algorithm 1):
> (1) MINs learn an inverse map under reweighted data distribution (Section 3.3)
> (2) MINs use specialized inference procedure to produce optimized $x$ values (Section 3.2).
> While Nguyen et al.’s method is clearly relevant and will be discussed in our paper, its relationship with MINs is similar to that of other conditional generative models, such as cGANs, which are already discussed.
>
> 3. cGAN baseline: In Appendix C.2, Figure 6, we present results from a cGAN baseline for the youngest face optimization task (compare to Figure 2). Note that a vanilla cGAN baseline tends to not perform optimization over the score value (i.e. doesn’t produce faces with small age), but rather produces images from the dataset distribution ignoring the score value. In general, static datasets could be highly skewed towards lower values of score that vanilla inverse-prediction might not be able to optimize score values.
>
> We are happy to clarify any further questions that the reviewer has. We hope that our response clearly presents the differences between our approach and Ngyuen et al.

---

### Author Response · Authors · 2019-11-15
**Author Response: Summary of Revisions**

We thank the reviewers for their constructive feedback. We have revised the paper to improve clarity and included quantitative results in the main text. We summarize the updates below.

Section 2:
1. Added reference to Ngyuen et.al. in related work

Section 3.2:
1. Added a probabilistic derivation of the inference scheme in Appendix A (newly added).

Section 3.3:
1. Added interpretation of the function $g$ in reweighting
2. Improved Section 3.3 by omitting the rearrangement of the loss term

Section 3.4:
1. Visualization of the randomized labeling scheme for a 1D function found here: https://ibb.co/album/i4d8qa (For reference, we used the same 1D function as https://tinyurl.com/min-1d-example )

Section 3.5:
1. Described the procedure for creating the augmented dataset in practice
2. Improved description of how reweighting (Section 3.3) is implemented in practice

Section 4:
1. Added self-contained description of batch contextual bandits task
2. Added an ablation of MINs w/o reweighting in Table 1
3. Added clear description to Figure 1 to describe how optimization is performed with MINs w/o inference (MIN - I); added a description of different rows in the caption of Figure 1.
4. Added quantitative score values to Figure 1
5. Added new tables (Table 2 and 3) with quantitative scores for face optimization task and MNIST inpainting
6. Added description of the architecture of $f^{-1}$ in Section 4.2 (protein fluorescence maximization) (previously in Appendix C.4)
7. Added self-contained description of the protein fluorescence maximization task, added a description of baselines

In all, we have quantitative results for five experiments, excluding benchmark functions, (Table 1, Figure 1, Table 2, Table 3, Table 5) and multiple ablations in Table 1 (w/o reweighting, w/o inference), Figure 1 (w/o reweighting, w/o inference), Figure 2 vs Figure 6, Appendix C.2 ( w/o reweighting), Table 5 (w/o reweighting, w/o inference), Table 4( w/o randomized labeling).

---

### Decision · Program_Chairs · 2019-12-19

**Decision:**

Reject

**Comment:**

This paper proposes Model Inversion Networks (MINs) to solve model optimization problems high-dimensional spaces. The paper received three reviews from experts working in this area. In a short review, R1 recommends Reject based on limited novelty compared to an ICDM 2019 paper. R2 recommends Weak Reject, identifying several strengths of the paper but also a number of concerns including unclear or missing technical explanations and need for some additional experiments (ablation studies). R3 recommends Weak Accept, giving the opinion that the idea the paper proposes is worthy of publication, but also identifying a number of weaknesses including a "rushed" experimental section that is missing details, need for additional quantitative experimental results, and some "ad hoc" parts of the formulation. The authors prepared responses that address many of these concerns, including a convincing argument that there is significant difference and novelty compared to the ICDM 2019. However, even if excluding R1's review, the reviews of R2 and R3 are borderline; the ACs read the paper and while they feel the work has significant merit, they agree with R2 and R3 that the paper needs additional work and another round of peer review to fully address R2 and R3's concerns.